# EmboMatrix: A Scalable Training-Ground for Embodied Decision-Making

## Abstract

**Embodied decision-making** enables agents to translate high-level goals into executable actions through continuous interactions within the physical world, forming a cornerstone of general-purpose embodied intelligence. Large language models (LLMs), with their general decision-making capabilities, offer a promising path to realize this potential; however, LLMs trained solely on language lack exposure to physical environments, limiting their true embodied understanding. To bridge this gap, we propose the concept of a **training ground**: a comprehensive infrastructure that provides task and scene simulation, embodied interaction, and feedback signals, offering a one-stop solution for LLM acquire genuine embodied decision-making skills. In this work, we present **EmboMatrix**, the first training ground of its kind, providing massive and diverse tasks with efficient simulation and precise rewards. EmboMatrix incorporates a series of novel techniques: a multi-agent data engine for large-scale task and scene generation, a distributed heterogeneous-hardware system for scalable simulation, and a multi-level reward architecture for precise supervision. Leveraging EmboMatrix, we cultivate **Embo-Brain**, an LLM whose embodied decision-making abilities emerge from extensive embodied interactions. Experiments show that EmboBrain-7B surpasses the 671B DeepSeek-R1 baseline by 9.5% on two challenging embodied decision-making benchmarks, demonstrating the power of interactive, environment-grounded learning for building truly intelligent embodied agents. The code will be released upon the paper's acceptance.

## 1 Introduction

**Embodied decision-making** (Li et al., 2024c) enables agents to translate high-level goals into executable actions through continuous interactions with the physical world, forming the cornerstone of general-purpose embodied intelligence. Without it, agents can hardly generalize across diverse tasks or operate effectively in complex, dynamic scenarios. Existing efforts largely follow two paradigms. End-to-end Vision-Language-Action (VLA) Kim et al. (2025) models map raw sensory inputs directly to low-level motor commands, implicitly performing embodied decision-making by integrating perception and action to achieve high-level goals, but they require vast imitation data and struggle with long-horizon planning. Hierarchical approaches Birr et al. (2024); Li et al. (2024b); Driess et al. (2023); Brohan et al. (2023); Mu et al. (2023); Wang et al. (2023a); Wu et al. (2023b); Wang et al. (2023c) decouple high-level reasoning from low-level control: low-level models execute primitive skills, while a high-level model orchestrates embodied decision-making by interpreting instructions, reasoning about the world, and decomposing tasks into actionable sub-goals. This structure simplifies long-horizon reasoning and allows integration of advanced models, such as large language models, into the decision-making loop. In this work, we adopt the hierarchical paradigm and focus on enhancing the high-level model's embodied decision-making to enable robust, generalizable, and adaptive behavior across diverse tasks.

Large language models (LLMs), with their advanced reasoning and general decision-making abilities, offer a promising foundation for embodied decision-making. Early work (Li et al., 2024c) demonstrated the zero-shot competence of LLMs on curated datasets. Some subsequent studies (Azzolini et al., 2025; Ji et al., 2025) enhanced these abilities through fine-tuning on specialized datasets, such as physically grounded question-answering pairs, injecting curated embodied knowledge. Although convenient, such non-interactive fine-tuning resembles a "brain in a vat": it promotes rote memorization rather than true understanding of physical dynamics, and the resulting gains are often limited

even with large amounts of training data. Achieving genuine mastery in embodied decision-making requires interactive learning, which involves acting, perceiving feedback, and adapting. However, direct training in the real world is costly, risky, and difficult to scale. This motivates the use of **high-fidelity simulation environments**, which replicate physical dynamics, enable efficient and large-scale interaction, and provide rich feedback, allowing agents to safely acquire and refine the skills needed for robust, real-world decision-making.

To enable genuine embodied decision-making, we introduce the concept of a **training ground**: a comprehensive infrastructure that provides task and scene simulation, realistic embodied interaction, and feedback signals, allowing models to acquire embodied decision-making skills through trial-and-error in physically grounded environments. Constructing such a training ground is highly challenging due to the inherent complexity of the embodied domain. At the data level, it requires generating a massive, diverse, and scalable curriculum of tasks that are both physically plausible and guaranteed to be solvable. At the system level, no existing simulation platform can support the high-throughput, large-scale interactions necessary to train high-capacity LLMs effectively. At the algorithmic level, it demands the design of an informative reward architecture tailored to embodied scenarios, providing dense supervision while avoiding the biases and limitations of manual reward engineering. Together, these technical challenges make the construction of a high-level training ground a complex and demanding task.

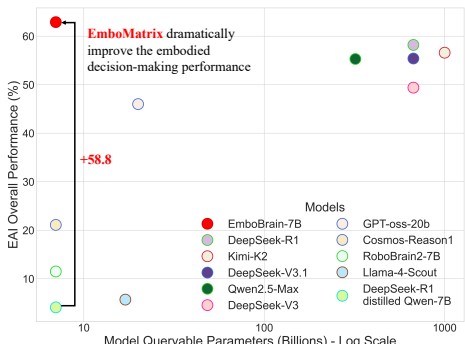

Figure 1: **EmboMatrix** substantially improves embodied decision-making, elevating a 7B base model's performance by +58.8 percentage points on the embodied agent interface enchmark, significantly outperforming both other domain-specialized models and much larger LLMs.

To this end, we introduce **EmboMatrix**, the first training ground designed for embodied decision-making, enabling high-throughput interaction and efficient training with massive and diverse tasks and scenes. EmboMatrix offers three key advantages: (i) **Data diversity**: a multi-agent driven automated data factory generates large numbers of tasks, enabling models to acquire generalizable capabilities across varied environments; (ii) **System scalability**: compatibility with distributed, heterogeneous hardware, combined with semantic abstraction and pre-caching of low-level processes, substantially improves simulation throughput; and (iii) **Informative supervision**: a hierarchical reward architecture delivers richer learning signals for embodied decision-making. Compared with conventional robot simulators, EmboMatrix provides a one-stop solution for embodied decision-making: it not only supports simulation of richer and more diverse scenarios, but also automatically generates physically grounded tasks and enables large-scale model training, resulting in substantially higher training efficiency and broader generalization.

Leveraging EmboMatrix, we train **EmboBrain**, an LLM whose embodied decision-making abilities emerge from extensive interaction within the training ground. This process effectively transforms a purely language-trained model into a truly embodied agent capable of perceiving, acting, and adapting in physical environments. Experiments show that EmboBrain-7B achieves substantial gains on multiple challenging embodied decision-making benchmarks, surpassing the strong DeepSeek-R1 baseline by 9.5% on average. These results demonstrate that interactive, environment-grounded learning, powered by a comprehensive training ground, is a transformative path toward building truly intelligent embodied agents.

## 2 RELATED WORK

**Embodied Decision Making**. LLM are used to generate text based action sequences directly Driess et al. (2023); Brohan et al. (2023); Mu et al. (2023); Wang et al. (2023a); Wu et al. (2023b); Wang et al. (2023c); translate instructions into executable code Liang et al. (2022); Singh et al. (2022); or provide intermediate representations consumed by downstream modules Jiang et al. (2022); Dalal et al. (2024). Knowledge-augmented variants improve grounding via dynamic memory Ding et al. (2023); Hazra et al. (2023); Li et al. (2024d); Chen et al. (2024). Reactive embodied agents combine LLM reasoning with online adaptation Birr et al. (2024); Tian et al. (2024); Liang et al. (2024). Additional work addresses anomaly detection, zero-shot knowledge extraction, and physics reasoning Sinha et al. (2024); Huang et al. (2022); Hao et al. (2023).

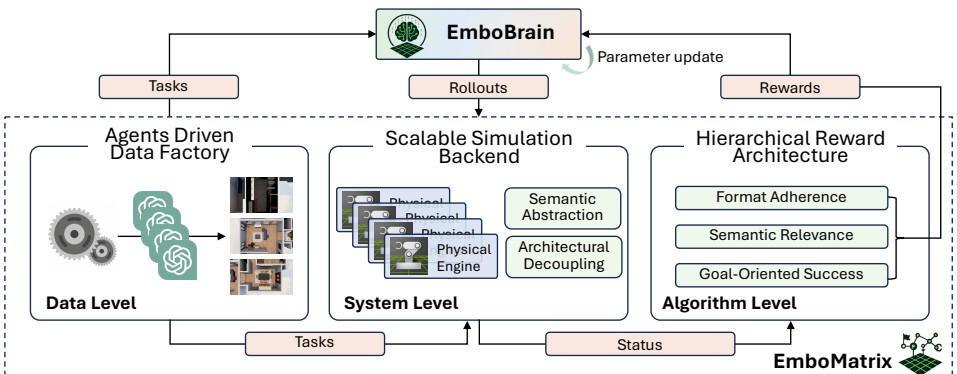

Figure 2: Overview of the *EmboMatrix* training pipeline.

**Simulator and Embodied Data Generation**. Recent benchmarks have expanded simulation for embodied agents, yet most—e.g., Behavior1K Li et al. (2024a), VirtualHome Puig et al. (2018), ALFRED Shridhar et al. (2020), iGibsonXia et al. (2020), Meta-World Yu et al. (2021), RLBench James et al. (2019), Habitat Szot et al. (2022), BEHAVIOR Srivastava et al. (2021), robosuite Zhu et al. (2025), TDW Transport Gan et al. (2021), SAPIEN Xiang et al. (2020), ManiSkill Mu et al. (2021); Gu et al. (2023), RFUniverse Fu et al. (2023), SoftGym Lin et al. (2021), EmbodiedBench Yang et al. (2025), and the unified platform RoboVerse Geng et al. (2025), focus on low-level manipulation. Tools such as ProcTHOR Deitke et al. (2022) broaden environments but remain rule-bound. Recent LLM and diffusion-aided methods (HOLODECK Yang et al. (2024c), ARCHITECT Wang et al. (2024), DiffuScene Tang et al. (2024), WorldCraft Liu et al. (2025b)) generate visually rich scenes yet ignore task constraints or object interactions. Meanwhile, embodied world models (DayDreamer Wu et al. (2023a), UniSim Yang et al. (2024a)) and RoboGen Wang et al. (2023b) focus on learning latent dynamics or simple layouts, often lacking explicit high-level semantic structures. A detailed comparison is shown in Tab 4. We address them with a multi-agent automated data factory that produces diverse, task-aware scenarios at scale.

**RL for Large Language Models**. LLM alignment via RL encompasses safety-aware feedback Dai et al. (2024); Lee et al. (2024); Chu et al. (2023), analyses of RLHF's effects on generalization and diversity Rafailov et al. (2023); Kirk et al. (2024), improved reward modeling via Nash learning and uncertainty estimation Munos et al. (2024), and more sample-efficient optimizers such as GPO and GRPO Zhao et al. (2024); Liu et al. (2025a). However, a systematic framework for training embodied decision making with RL remains lacking, limiting progress toward grounded embodied decision-making in complex environments. The work most closely related to ours is Fei et al. (2025), which introduces RL post-training for embodied decision-making. In contrast, we concretely instantiate a physics–based simulation system, provide substantially larger-scale data, and design a more informative reward system tailored to embodied decision-making.

## 3 PROBLEM FORMULATION

Let $\mathbf{B}_\theta$ be a model for high-level embodied decision-making parameterized by learnable weights $\theta$. Given a high-level instruction $I$ and the current embodied scene $S$ as input, the action sequence $\mathbf{a}$ in a physical space is then obtained as

$$\mathbf{a} = \mathbf{B}_\theta(S, I), \quad \text{where } \mathbf{a} = (a_1, \ldots, a_H) \text{ and } a_i \in \mathcal{A}, \tag{1}$$

where $\mathcal{A}$ is a predefined skill library. This process is defined as **embodied decision-making**.

Consider a **training ground** cultivates such a embodied decision-making model. Mathematically, let $\mathcal{F}_{\mathcal{D}_{\text{task}}, \mathcal{M}, \mathcal{R}}(\cdot)$ be a training ground parameterized by a collection of embodied tasks $\mathcal{D}_{\text{task}}$, an interactive physical simulator $\mathcal{M}$, and a reward architecture $\mathcal{R}$. Here an embodied task set $\mathcal{D}_{\text{task}} = \{T_i\}_{i=1}^N$ has $N$ tasks, where $T_i = (S_i, I_i, \mathcal{G}_i)$ is the $i$th tasks with $S_i$, the configuration that instantiates the scene in the simulator, containing all task-relevant objects and their initial states, $I_i$, the high-level instruction of $T_i$ (e.g. heat sandwich) and $\mathcal{G}_i$, the target conditions by binding them to concrete assets (e.g., *sandwich inside microwave*).

Let $\mathbf{B}_{\theta_0}$ be an input base model. Then, the optimized embodied decision-making model is obtained as

$$\mathbf{B}_{\theta^*} = \mathcal{F}_{\mathcal{D}_{\text{task}}, \mathcal{M}, \mathcal{R}}(\mathbf{B}_{\theta_0}). \tag{2}$$

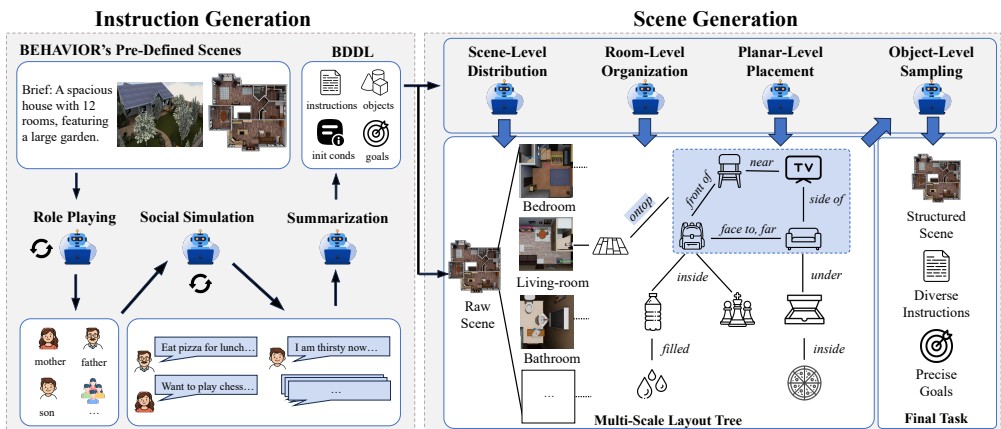

Figure 3: **Pipeline of Multi-Agent-Driven Automated Data Factory.**

The optimization objective is to find the optimal parameters $\theta^*$ by maximizing the expected reward across this task distribution:

$$\theta^* = \arg \max_\theta \mathbb{E}_{T \sim \mathcal{D}_{\text{task}}} \left[ \mathcal{R} \left( \mathcal{M} \left( T, \mathbf{B}_\theta(S, I) \right) \right) \right], \tag{3}$$

In this work, our proposed system, **EmboMatrix**, is the concrete realization of the training ground $\mathcal{F}_{\mathcal{D}_{\text{task}}, \mathcal{M}, \mathcal{R}}(\cdot)$, which will be described in Section 4.

## 4 METHODS

The preceding section established that enhancing an agent's embodied decision-making capability requires a comprehensive **training ground**. In this chapter, we introduce **EmboMatrix**, which operates as a system to continuously improve an agent's capabilities of embodied decsion-making. As shown in Figure 2, the entire training process is orchestrated by EmboMatrix. The cycle begins at the Data Level, where our agents driven data factory procedurally generates a diverse tasks set. This task is presented to the models, which produces a action sequence rollout. The action sequence is then executed at the System Level by our Scalable Simulation Backend, which leverages a high-fidelity physical engine to produce the final environments status for the action sequences. Finally, at the Algorithm Level, our hierarchical reward architecture evaluates this final environments status. This reward signal drives the parameter update for the EmboBrain, completing the learning loop. The following sections will detail the design of each of these three synergistic components.

### 4.1 MULTI-AGENT–DRIVEN AUTOMATED DATA FACTORY.

**Multi-agent social simulation for instruction generation.** Since generating one task amounts to producing $(\mathcal{S}, \mathcal{I}, \mathcal{G})$, we bootstrap from BEHAVIOR's pre-defined scenes (e.g., hotel rooms, household interiors), yielding a base layout $\mathcal{S}_0$ with plausibly distributed asset states (Li et al., 2024a). To synthesize diverse and scene-faithful tasks, as shown in Fig 3, we employ a multi-agent social simulation module with role-playing (Tang et al., 2025; Li et al., 2023). The module first extracts information from $\mathcal{S}_0$, renders RGB images from egocentric and top-down views, and prompts a VLM to distill a concise textual scene description. Socially simulated agents then generate relevant characters (e.g., *father*, *mother*, *child*, *home robot*) and conduct multi-round dialogues in which human agents articulate language-based needs for the embodied agent; the exchanges are iteratively refined until concrete requirements emerge. Each scene can be repeatedly simulated to create a large number of semantically diverse instructions $\mathcal{I}$. An LLM-based agent subsequently summarizes $\mathcal{I}$ into a BDDL format (Li et al., 2022), which contains $\mathcal{G}$, $\mathcal{O}$, and $\mathcal{IC}$.

**Multi-level scene generation.** We further employ a multi-agent and multi-level scene generation module to efficiently and faithfully transform $\mathcal{S}_0$ into an instantiated scene $\mathcal{S}$. As shown in Fig 3, these agents allocate $\mathcal{O}$ across different rooms, analyze the states of each room, and design a **multi-scale layout tree** such that object placement not only satisfies $\mathcal{IC}$ but also ensures spatial plausibility and visual aesthetics. Beyond basic relations (e.g., *under*, *inside*), the tree refines specializations such as *ontop* (Yang et al., 2024b), while employing constraints like *faceto* and *insideof* to regulate object placement on the same plane with higher precision and consistency. Finally, sampling agents place

objects in the simulator according to the tree structure, resulting in an executable task $\mathcal{T}$ with a realistic and well-structured scene.

With this multi-agent–driven data factory, we can randomize across scenes and social simulations, generating realistic and diverse tasks and laying a solid data foundation for subsequent training.

### 4.2 Scalable Simulation Backend

To facilitate the massive scale of interaction required by our learning objective in equation 3, the system architecture must overcome two fundamental hurdles: the computational cost of individual physical interactions and the systemic overhead of massively parallel execution. We address these through a principled approach of semantic abstraction and architectural decoupling.

**Semantic Abstraction via a Pre-Cached Physics Interface.** A primary performance bottleneck is the *granularity mismatch* between the LLM's high-level semantic commands and the simulator's computationally expensive, low-level micro-dynamics. To resolve this, our **pre-cached language-physics interface** acts as a semantic abstraction layer. For common interactions, instead of simulating the full physical process, we bypass the costly dynamics by directly instantiating a valid, physically plausible outcome from a pre-computed set of post-conditions. This outcome-based simulation approach dramatically accelerates throughput while preserving the semantic consequences crucial for the learning signal. See more details in Appendix B.

**Architectural Decoupling for Massively Parallel Rollouts.** A second challenge arises from the conflicting resource requirements of LLM training and large-scale physics simulation, which renders a monolithic architecture inefficient and unscalable. We resolve this by employing an architecturally decoupled, distributed simulation backend. This service-oriented design separates the LLM trainer from a heterogeneous pool of simulation workers, allowing each component to run on specialized, optimal hardware. However, this distributed architecture necessitates sophisticated scheduling to manage communication and I/O overheads. We address this with two key components: a **Resource-Scheduler** that predictively pre-loads future scenes onto idle workers to hide latency, and a **Task-Dispatcher** that maps incoming action sequences to these pre-warmed simulators to maximize utilization (details in Appendix B). This design resolves systemic resource conflicts and enables high-throughput rollouts at a scale unattainable by conventional approaches.

### 4.3 Hierarchical Reward Architecture

A central challenge in applying RL to long-horizon embodied tasks is the severe credit assignment problem. A sparse, binary reward for final task completion provides insufficient guidance for meaningful exploration in a combinatorially large state-action space. To overcome this, we eschew a single, static reward function and instead introduce a **hierarchical reward architecture**. This architecture provides a dynamic, multi-stage curriculum of supervision signals designed to guide the agents from basic format adherence to complex, goal-oriented semantic reasoning. The total reward $r_i = r_f + r_r + r_g$ is composed of three tiers, each targeting a distinct stage of the learning process.

**Format Adherence ($r_f$).** The foundational stage of the curriculum focuses on teaching the agents to generate well-formed outputs. The agents receives a binary format reward, $r_f$, which provides a simple, rule-based signal for whether its output conforms to the prescribed action schema. As demonstrated in prior work (Wang et al., 2025; Shao et al., 2024), this initial phase of enforcing syntactic correctness is crucial for stabilizing early training and ensuring a high rate of executable rollouts.

**Semantic Relevance ($r_r$).** Once the agent can generate valid syntax, the curriculum shifts to guiding exploration. The relevance reward, $r_r$, serves as a dense, intermediate signal that bridges the gap between random exploration and goal-directed behavior. It is defined as:

$$r_r \;=\; \beta \,\big|\, \mathcal{O}_{\text{goal}} \,\cap\, \mathcal{O}_{\mathbf{a}} \big|, \qquad \beta > 0,$$

where $\mathcal{O}_{\text{goal}}$ is the set of objects required to achieve the goal and $\mathcal{O}_{\mathbf{a}}$ is the set of unique objects the agent's action sequence interacts with. By rewarding the agent for interacting with goal-relevant objects, $r_r$ effectively shapes the agent's behavior, encouraging it to focus its exploration on the semantically pertinent subset of the vast interaction space.

**Goal-Oriented Success ($r_g$).** The final tier of the curriculum provides the ground-truth signal for task completion. The goal reward, $r_g$, is a sum over the individual goal predicates defined by the task:

$$r_g \;=\; \alpha \sum_{g_k \in \mathcal{G}} \mathbb{I}\big[g_k(s_H) = 1\big], \qquad \alpha > 0.$$

Table 1: Comprehensive comparison of model success rates (%) on two benchmarks: our agent-generated benchmark and the Embodied Agent Interface (EAI) benchmark. On our agent-generated benchmark, our model, **EmboBrain-7B**, outperforms GPT-4o and DeepSeek-R1 by 18.2% and 14.8%, respectively. Models are grouped into three categories: cells with ☐ denote large-parameter models, ☐ denote embodied-scene enhanced models, and ☐ denote EmboBrain-related models.

| Model | Size | Our Agent-Generated Benchmark | | | | | Embodied Agent Interface (EAI) Benchmark | | | | |
|---|---|---|---|---|---|---|---|---|---|---|---|
| | | Overall | Pick and Place | Appliances Using | Kitchen Operation | Compound Task | Overall | Pick and Place | Appliances Using | Kitchen Operation | Compound Task |
| GPT-4o-mini | - | 12.0 | 17.2 | 13.4 | 6.5 | 15.1 | 26.5 | 43.5 | 27.1 | 8.1 | 21.7 |
| GPT-o1-mini | - | 33.4 | 50.2 | 39.6 | 17.2 | 38.6 | 58.6 | 66.9 | 70.0 | 26.5 | 60.9 |
| GPT-4o | - | 45.0 | 45.9 | 43.8 | 38.1 | 55.6 | 44.8 | 62.2 | 58.0 | 22.3 | 37.6 |
| GPT-oss-20b | 20B | 20.6 | 27.6 | 27.5 | 9.2 | 26.7 | 46.0 | 60.6 | 45.0 | 35.4 | 40.6 |
| DeepSeek-V3 | 671B | 40.1 | 59.3 | 38.6 | 19.7 | 56.0 | 49.4 | 52.0 | 62.8 | 29.5 | 50.5 |
| DeepSeek-R1 | 671B | 51.6 | 67.1 | 56.1 | 36.6 | 58.7 | 58.2 | 61.8 | 79.5 | 32.0 | 58.6 |
| DeepSeek-V3.1 | 671B | 41.0 | 48.6 | 44.2 | 27.2 | 53.1 | 55.4 | 63.9 | 64.7 | 42.4 | 51.8 |
| Llama-4-Scout | 17B | 1.4 | 5.0 | 1.1 | 0.3 | 0.2 | 5.7 | 8.7 | 4.1 | 3.2 | 5.2 |
| Kimi-K2 | 1T | 48.5 | 52.1 | 47.2 | 41.7 | 56.8 | 56.6 | 62.2 | 68.4 | 34.6 | 57.0 |
| Qwen2.5-Max | 320B | 11.7 | 16.7 | 16.7 | 3.6 | 16.3 | 55.3 | 63.5 | 69.4 | 33.0 | 53.5 |
| RoboBrain2-7B | 7B | 20.9 | 26.0 | 22.2 | 18.5 | 19.1 | 11.5 | 12.6 | 17.3 | 6.7 | 10.8 |
| Cosmos-Reason1 | 7B | 5.9 | 7.9 | 7.8 | 4.0 | 5.7 | 21.1 | 21.7 | 25.9 | 14.4 | 21.7 |
| 1.5B Base | 1.5B | 3.8 | 4.8 | 2.8 | 0.9 | 8.4 | 0.2 | 0.1 | 0.1 | 0.0 | 0.3 |
| **EmboBrain-1.5B** | 1.5B | 48.7 | 54.8 | 47.2 | 44.1 | 51.4 | 8.9 | 8.2 | 7.7 | 7.2 | 10.3 |
| 7B Base | 7B | 5.5 | 11.9 | 5.6 | 2.7 | 14.6 | 4.1 | 1.0 | 4.9 | 1.3 | 6.8 |
| **EmboBrain-7B** | 7B | **65.8** | **73.2** | **62.7** | **60.1** | **70.3** | **62.9** | **75.6** | **70.0** | **37.7** | **61.4** |

**Task: Heat chicken**

```
GPT-4o: score 0

move("chicken wing")
pickup("chicken wing")
move("microwave")          Forget open
toggle_on("microwave")     microwave
cook("chicken wing", "heat")
move("table")
place("chicken wing", "ontop", "table")
```

```
DeepSeek-R1: score 0.5

move("chicken wing")
pickup("chicken wing")
move("microwave")          Forget toggle_on
open("microwave")          microwave
place("microwave", "inside")
close("microwave")
cook("chicken wing", "heat")
```

```
EmboBrain 7B Model: score 1

move("chicken wing")
pickup("chicken wing")
move("microwave")
open("microwave")          Correct
place("microwave", "inside")
close("microwave")
toggle_on("microwave")
```

Figure 4: Qualitative comparison on the *Heat Chicken* task. GPT-4o omits opening the microwave door; DeepSeek-R1 inserts the food but never toggles the appliance on; only our EmboBrain-7B produces a complete, executable sequence and succeeds in simulation.

The multi-level reward system ensures that, on the one hand, informative guidance is provided during the early stages of model training, while on the other hand, the goal achievement serves as the primary driver, preventing the model from falling into inductive bias.

## 5 EXPERIMENTS

In this section, we present comprehensive experiments designed to validate the efficacy of the **EmboMatrix** framework . Our evaluation is structured to answer three major questions, which directly correspond to the data, system, and algorithmic challenges addressed in this work:

1. **EmboMatrix Effectiveness:** Does end-to-end training within the EmboMatrix substantially improve the performance of different LLMs on long-horizon embodied decision-making tasks?
2. **EmboMatrix Scalability:** Do the data and system components of EmboMatrix deliver the requisite diversity and throughput to support scalable, long-term training?
3. **Algorithmic Efficiency:** How effective is our hierarchical reward architecture at improving sample efficiency and overall learning outcomes compared to simpler, baseline reward schemes?

### 5.1 OVERALL PERFORMANCE ON EMBODIED DECISION-MAKING

We begin by addressing our first question: to what extent does end-to-end training within the **EmboMatrix** framework improve the performance of LLMs on complex embodied tasks? To this end, we train our models and evaluate them on two challenging benchmarks.

**Experimental Setup.** We train two models, denoted **EmboBrain-1.5B** and **EmboBrain-7B**. These models are initialized from the publicly available DeepSeek-R1 distilled Qwen-1.5B and 7B checkpoints (referred to as "1.5B base and 7B base" in table 1)., respectively, and are subsequently trained within our **EmboMatrix** framework using the GRPO algorithm. The input prompt provided to the model at each decision step comprises four key components: a predefined task description, a profile of the agent's capabilities, a simulator-generated description of the current scene state, and a formatting instruction for the output. LLM optimization ran on an $8 \times$ A100 cluster, while parallel

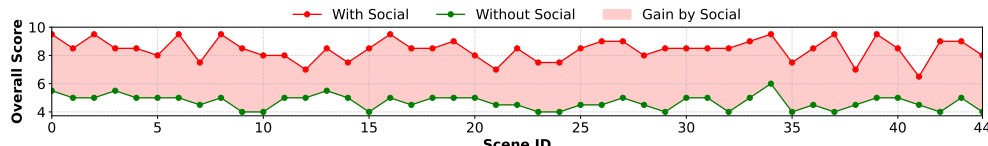

Figure 5: Social simulation significantly increases the diversity of tasks.

simulation was executed on a pool of 16 graphics GPU. The training dataset for our models is entirely procedurally generated by our Multi-Agent–Driven Automated Task Factory. This process leverages a base set of 45 diverse scenes from the Behavior-1K environment (Li et al., 2024a) to synthesize a large-scale training corpus. For evaluation, we assess model performance on two distinct, held-out benchmarks, each containing 100 challenging task-scene pairs: First, **Internal-Verified**: A manually verified test set generated by our own pipeline to ensure high-quality and challenging scenarios. Second, **Embodied Agent Interface (EAI) Benchmark**: a published benchmark for embodied decision-making based on the BEHAVIOR dataset (Li et al., 2024c). Specifically, we focus on the **Action Sequencing** track. We adopt this setting because it necessitates dynamic environment interaction and step-by-step grounding, thereby serving as the most direct metric for our core goal: enhancing embodied decision-making through active world modeling, as opposed to static semantic parsing tasks. All tasks are categorized into four types: (1) *Pick and Place*, involving basic object manipulation and organization; (2) *Appliance Usage*, requiring the agent to change the state of household devices; (3) *Kitchen Operations*, including food-related tasks such as heating or freezing; and (4) *Compound Tasks*, which consist of multi-step instructions spanning multiple categories.

**Results**. We compare our models with representative proprietary and open-source LLMs with well-accepted leading performance. To reduce sampling variance, each model is queried ten times per task, and we report the average result for each pair. Table 1 summarizes the outcomes and highlights key observations: (i) Physical interaction training with **EmboMatrix** significantly enhances the embodied decision-making capabilities of general-purpose LLMs. Specifically, for 1.5B model, EmboMatrix achieves performance improvements of 44.9% and 8.7% for the 7B model, and 60.3% and 58.8%, respectively, across the two benchmarks. The 1.5B model exhibits relative limited performance on the EAI benchmark constrained by the capabilities of base model. (ii) Post-training with **EmboMatrix** enables a 7B model to outperform much larger models, including the proprietary GPT-4o and the 671B-parameter DeepSeek-R1 on both embodied decision-making tasks. (iii) The largest gains are observed in the most challenging category, Kitchen Operation, indicating that physically grounded learning is particularly beneficial for complex, multi-step reasoning.

Figure 4 compares roll-outs for the *Heat Chicken* task. All three models GPT-4o, DeepSeek-R1, and our EmboBrain-7B produce seemingly reasonable action sequences, yet only our model completes the task in simulation. GPT-4o omits opening the microwave door before place the chicken; DeepSeek-R1 places the food correctly but never toggle on. In contrast, EmboBrain-7B executes the full sequence: open door, insert food, close door, power on—and therefore succeeds. This example illustrates that action sequences appearing valid in text may still fail in physical environments, underscoring the importance of interaction in physical environments.

## 5.2 SCALABILITY OF EMBOMATRIX.

### 5.2.1 DATA DIVERSITY AND QUALITY.

**Experimental Setup.** We conduct two ablation studies focusing on task diversity and scene quality. First, to assess **task diversity**, we compare our social simulation-based approach against a direct generation baseline. For 45 scenes from behavior-1k, we generate 10 tasks with each method and use a GPT-4 to score the resulting diversity (details in Tab.6). Second, to evaluate **scene quality**, we compare our multi-level lay-

Table 2: Multi-level layout tree significantly improves the quality of scene generation.

| Method | Generation Rate | Aesthetic Score | Verification Pass Rate |
|---|---|---|---|
| W/o Tree | 49.29 | 7.30 | 47.83 |
| LayoutGPT | 45.72 | 7.22 | 25.00 |
| Our | **71.43** | **8.11** | **98.00** |

out tree against two baselines (behavior's built-in functions and LayoutGPT Feng et al. (2023)) on 140 tasks. We measure performance across four metrics: generation success rate, time, verification pass rate, and scene aesthetics, with the latter also scored by a GPT-4 evaluator (details in Tab.7).

**Social Simulation Enhances Task Diversity.** As shown in Fig. 5, incorporating social simulation significantly increases task diversity, achieving an average score of 8.42, compared to 4.70 without

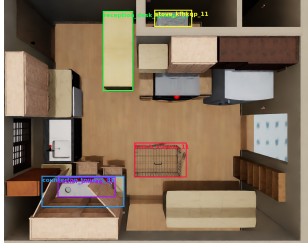 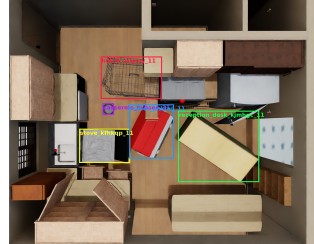

(a) Multi-level layout tree        (b) No multi-level layout tree        (c) LayoutGPT

Figure 6: Multi-level layout tree significantly improves the aesthetics of scene generation.

social simulation. Examples are provided in Tab. 5. **Multi-level Layout Tree Improves Scene Generation Quality.** Quantitative results in Tab. 2 show that the multi-level layout tree achieves the highest scene generation rate (71.43%) and verification pass rate (98.00%). Failures result from limited space or unmet task conditions. Without the layout tree, unstructured placements, such as positioning a bag before the table, lead to misalignments, reducing the generation rate to 49.29%. The absence of pre-checks for feasible robot positions also decreases the pass rate. LayoutGPT performs worst due to limited spatial reasoning, causing cluttered room layouts. Scenes with the layout tree also achieve the highest aesthetics score (8.11), while those generated by other methods often miss key objects, place them incorrectly, or result in cluttered layouts, as shown in Fig. 5.

### 5.2.2 SIMULATION EFFICIENCY SUPPORT TRAINING PROCESS.

Table 3 presents an ablation study validating our system's efficiency by measuring the average per-rollout simulation latency. By progressively enabling three key optimizations—**Pre-cached Execution**, a **Resource Scheduler**, and a **Task Dispatcher**—our full system achieves a nearly 50-fold reduction in overhead. This significant improvement is critical, providing the massive simulation throughput required for our large-scale RL experiments.

Table 3: Abalation of efficiency. Latency measures the average time to simulate a scenario. We reduces this overhead by over 50×.

| Pre-cached Execution | Resource Scheduler | Task Dispatcher | Latency (s) |
|:---:|:---:|:---:|:---:|
| | | | 3.48 |
| ✓ | | | 0.85 |
| ✓ | ✓ | | 0.14 |
| ✓ | ✓ | ✓ | **0.07** |

### 5.3 HIERARCHICAL REWARD ARCHITECTURE IMPROVE TRAINING EFFICIENCY

To validate the effectiveness of our hierarchical reward architecture, we conduct an ablation study on the impact of the semantic relevance reward ($r_r$). As illustrated in Figure 7, the learning curves demonstrate a stark difference in training efficiency and final performance. The agent trained without the semantic relevance reward makes minimal progress, stagnating at a low goal-oriented success reward. This indicates that without this intermediate guidance, the agent is unable to solve the credit assignment problem and discover a successful policy in a sparse-reward setting. In contrast, the agent trained with the semantic

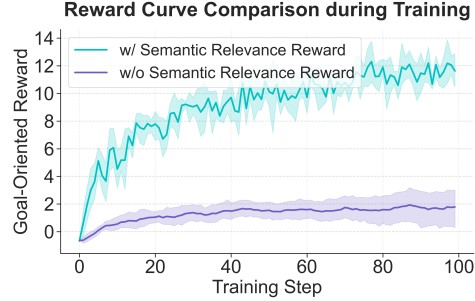

Figure 7: Semantic relevance reward effect.

relevance reward exhibits a rapid and stable increase in performance, achieving a significantly higher final reward. This result confirms that semantic relevance reward is a critical component of our training ground's design, enabling efficient learning by providing a dense and meaningful signal that effectively guides the agent's exploration towards goal-oriented behaviors.

## 6 CONCLUSION

**Conclusion.** In this work, we argue for a paradigm shift from learning language data to interactive learning for embodied agents. We introduced **EmboMatrix**, the first scalable **training ground** that makes this vision practical by addressing the challenges of data, system, and algorithm design. Our framework successfully transforms general-purpose Large Language Models into powerful **EmboBrain** models, with our EmboBrain-7B achieving a 14.2% performance gain over a strong baseline. EmboMatrix provides a complete and validated solution for the continuous improvement of embodied agents through direct, simulated experience.

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

# A    DETAILED EXPLAINTION OF OUR MULTI-AGENT DATA FACTORY

Our multi-agent-driven automated data factory is designed to generate a vast and diverse set of realistic, long-horizon embodied AI tasks. The entire pipeline can be broken down into two primary stages: first, we leverage multi-agent social simulation to generate semantically rich task instructions; second, we employ a multi-level scene generation process to construct the corresponding physically interactive 3D environments. This section provides a detailed walkthrough of each component.

## A.1    STAGE 1: SOCIAL SIMULATION FOR TASK INSTRUCTION GENERATION

To avoid generating homogeneous and short-term tasks, we adopt a strategy inspired by MATRIX, performing social simulations within pre-existing embodied scenes to generate meaningful instructions.

The process begins by preprocessing 45 diverse scenes from the Omnigibson simulation platform. For each scene, we extract key information, including scene images and textual descriptions of room types and unique environmental attributes. This information is fed into a **Role Playing Agent**, which generates plausible character profiles tailored to the scene. For instance, in a house environment, it might simulate a family, defining each member with attributes like name, age, job, relationships, and hobbies.

Next, a **Social Simulation Agent** receives both the character profiles and detailed scene information (e.g., room names and sizes). It then produces tasks that a robot might perform to assist the characters within that social context. A typical generated task might be: "Bring the chess from the counter in the living-room to the table in the garden for dad."

Finally, to make these language-based instructions machine-executable, a **Summarization Agent** maps the semantic task into the structured BDDL format. This format explicitly defines the goal conditions for the task. During a post-processing phase, we match the objects involved in the task with available 3D assets and use a combination of rule-based filters and a validation agent to eliminate unsupported or duplicate tasks, ensuring the quality and feasibility of the generated data.

## A.2    STAGE 2: MULTI-LEVEL SCENE GENERATION

Once a task is defined in BDDL, we need to generate a 3D scene where the initial conditions are met and the task is physically executable. To automate the construction of diverse, multi-room scenes for these long-horizon tasks, we designed an interpretable, multi-level generation framework. The process unfolds across three hierarchical levels: scene, room, and planar.

**Scene Level: Object Distribution.**    To mitigate the complexity of generating a full multi-room scene, we first operate at the scene level. We designed a rule-based **Scene-level Distribution Agent** that takes all objects and initial conditions specified by the task, along with the room layout of a base scene (e.g., room names and sizes). This agent assigns each object and its corresponding state conditions to an appropriate room. The distributor considers task requirements (e.g., if two objects must start on the same table, they are assigned to the same room) while also balancing scene richness and common sense. It prioritizes larger rooms but also considers room function to avoid unnatural clustering or excessive dispersion of objects. After this stage, all objects are assigned to a room and are ready for precise placement.

**Room Level: Layout Tree.**    Within each room, we use a multi-level tree structure to represent the object layout at a coarse granularity. This tree contains two node types: object nodes and relation nodes. The relations are based on nine object state functions supported by Omnigibson (e.g., *ontop*, *under*, *inside*). If two objects share a spatial relationship, they are connected via a relation node. The tree is built sequentially, with the root node typically being the room's floor. A **Room-level Organization Agent** is responsible for creating this structure. It takes the list of objects assigned to the room and a top-down view as input, then outputs a layout tree based on common sense, object sizes, and their approximate positions. For example, it knows a small backpack is more likely to be *ontop* of a table than on the floor. The agent prioritizes satisfying the task's initial conditions and iteratively refines the tree. If a generated layout proves invalid during placement, the incorrect relation is fed back to the agent as historical context to prevent similar errors in subsequent attempts.

**Planar Level: Object constraints in the same plane.** This level addresses the challenge of fine-grained placement, particularly for the *ontop* relation. Placing multiple objects on a single surface (like a floor or a large table) using random sampling often results in cluttered, aesthetically unpleasing arrangements that can block a robot's path.

To solve this, we employ a **Planar-Level Placement Generator**, inspired by recent work, which operates at a finer granularity. For objects placed on the same base surface, this agent establishes precise spatial constraints between them, such as *faceto*, *nextto*, or *alignedwith*. For example, it might specify that a newly placed chair should be *in front of* the desk and *face to* the computer monitor. The optimal position for a new object is found by discretizing the base surface into a grid and selecting the grid cell that maximizes a score derived from these constraints.

Throughout this optimization, two hard constraints are strictly enforced: (1) the new object must not collide with any existing objects, and (2) there must be a valid nearby position for the robot to be spawned, ensuring the object is interactable.

**Object Level: Object Placement Sampling.** At this level, we use the APIs provided by the Omnigibson system for various relations to sample object positions. This ensures that objects do not collide with other items in the scene while satisfying the positional constraints specified at the Room and Planar levels. The entire implementation is integrated into our **Object-level Sampling Agent**.

After processing all rooms, the complete embodied scene is saved. Finally, we perform a validation step by loading the scene and verifying that a robot can successfully manipulate the task-relevant objects, confirming the task's executability.

## B    DETAILED EXPLAINTION OF OUR SCALABLE SIMULATION BACKEND

This appendix provides a detailed technical breakdown of the **Scalable Simulation Backend**, a core component of EmboMatrix. As discussed in the main text, our approach is built on two fundamental principles designed to address the primary bottlenecks in large-scale interactive learning: **semantic abstraction** to accelerate individual physical interactions, and **architectural decoupling** to enable massively parallel rollouts.

The principle of semantic abstraction is concretely implemented through our **Pre-Cached Language-Physics Interface** (detailed in Sec. B.1). This component is responsible for grounding the LLM's high-level action sequences into low-latency physical state transitions.

Similarly, architectural decoupling is realized by the **Distributed Simulation Backend** (detailed in Sec. B.2). This service-oriented system, composed of a central manager and a fleet of heterogeneous worker nodes, is engineered to manage parallel simulations efficiently and hide system-level overheads.

The following subsections will detail the design and implementation of each of these core components.

### B.1    SEMANTIC ABSTRACTION: THE PRE-CACHED LANGUAGE-PHYSICS INTERFACE

A primary performance bottleneck in high-fidelity simulation is the *granularity mismatch* between the LLM's high-level semantic commands (e.g., `place(apple, table)`) and the simulator's computationally expensive, low-level micro-dynamics (e.g., contact forces, friction). To resolve this, we designed a structured interface that grounds the LLM's language outputs into executable state changes in the simulator.

Following a "language in, language out" paradigm, the agent generates a complete program $\pi = [a_1, \ldots, a_H]$, where each step $a_i$ is a structured token tuple, `<skill, arg`$_1$`, ..., arg`$_k$`>`, drawn from a predefined skill library. This design retains the generality of token-level inference while enabling physical grounding through symbolic decomposition. An execution engine interprets this action sequence step-by-step, triggering corresponding low-level controllers.

Crucially, to accelerate this process, we introduce a **pre-cached, outcome-based simulation** mechanism. For common interaction skills whose outcomes are quasi-static (e.g., placing an object), we pre-compute a manifold of valid and physically plausible terminal poses during an offline scene analysis phase. At runtime, once the skill's preconditions are met (e.g., the robot holds the object

near the target), the system bypasses the costly continuous motion simulation and collision resolution entirely. Instead, it directly instantiates a valid outcome from the pre-cached set. This approach preserves the crucial semantic consequences required for the reward signal while accelerating individual skill execution by an estimated **5x to 100x**, making large-scale, closed-loop training in complex environments computationally feasible.

### B.2    ARCHITECTURAL DECOUPLING: THE DISTRIBUTED SIMULATION SYSTEM

The second major challenge stems from the conflicting resource requirements of LLM training (typically compute-bound) and large-scale physics simulation (often memory- and graphics-bound). A monolithic architecture that co-locates these processes is inherently inefficient and unscalable. We resolve this by employing an **architecturally decoupled, distributed simulation backend**, as illustrated in Figure 8. This service-oriented design separates the LLM trainer from a heterogeneous pool of simulation workers, allowing each component to run on its specialized, optimal hardware.

This distributed system is composed of three key parts that work in concert to hide latency and maximize throughput.

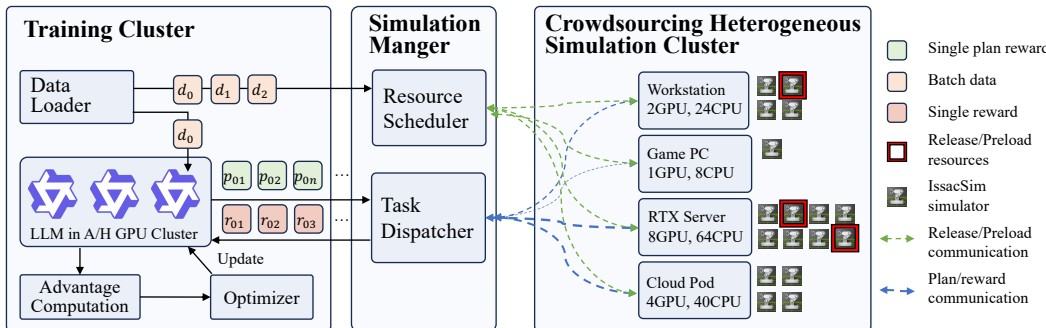

Figure 8: **The Architecture of the Distributed Simulation Backend.** The LLM trainer generates rollout action sequences and updates its policy via reinforcement learning. A central Simulation Manager schedules these action sequences across a heterogeneous, crowdsourced fleet of simulators. The system strategically overlaps scene loading with execution and streams the resulting experiences (observations, rewards) back to the learner, maximizing hardware utilization.

**Heterogeneous Simulator Cluster.**    A central simulation manager coordinates a distributed pool of worker nodes. These can be dedicated servers, workstations, cloud VMs, or even idle commodity PCs. Each node runs a lightweight daemon that self-registers with the manager, reports its available resources (GPU, VRAM, CPU), and fetches compressed simulation scenarios on demand. Because the simulators operate as independent services requiring no inter-node communication, the system can efficiently leverage globally distributed, commodity hardware, scaling rollout capacity without needing expensive, high-speed interconnects.

**Resource-Scheduler.**    To mitigate the significant latency caused by loading complex scenes, the scheduler acts proactively. At any given training step $t$, the scheduler polls all worker nodes and inspects the LLM's dataloader to predict the scenarios needed for future steps $t + 1, \ldots, t + k$. It then pre-assigns these future scenarios to idle nodes, which begin pre-loading the required assets. By the time a rollout is dispatched for execution, the corresponding simulation environment is already initialized and "pre-warmed," nearly eliminating loading stalls from the critical path of the training loop.

**Task-Dispatcher.**    Once the agent produces a batch of action sequences, the dispatcher maps each action sequence to its corresponding pre-warmed simulator slot on a worker node. It launches the executions in parallel and streams the resulting experiences back to the learner. As soon as a simulation slot becomes free, it is immediately reused for the next queued action sequence, ensuring that the simulation hardware remains continuously saturated. This combination of proactive scheduling and dynamic dispatching ensures that wall-clock training time is dominated by the necessary physics

Table 4: Comparison of our work with previous works that can generate interactive 3D scenes and train agents with embodied tasks in 6 aspects. Here, **Fully Automated** refers to the entire process, from embodied task generation to scene generation and RL of the agent, being completed without any human intervention. **Tasks with MASS** means the tasks are generated via multi-agent social simulation frameworks. **Scenes from tasks** means the ability to generate executable embodied scenes for any given task. **Multi room** means tasks executed in a multi-room embodied scene. **Interpretable** means the scene generation process is interpretable. **Self-Verifying** means the generated scene will be verified for completion feasibility.

| Methods | Fully Automated | Tasks with MASS | Scenes from tasks | Multi-room | Interpretable | Self-Verifying |
|---|---|---|---|---|---|---|
| Behavior-1k | | | | ✓ | ✓ | |
| ProcTHOR | | | | ✓ | ✓ | |
| Holodeck | | | | ✓ | ✓ | |
| Architect | | | | ✓ | | |
| RoboGen | ✓ | | ✓ | | ✓ | |
| Ours | ✓ | ✓ | ✓ | ✓ | ✓ | ✓ |

simulation itself, rather than by queueing or I/O bottlenecks. This design is what makes our large-scale, physically-grounded LLM training practical, with system overhead accounting for only about 20% of the total training time.

## C    DETAILS OF LEARNING METHODS

We train the high-level embodied decision-making model with *Group Relative Policy Optimization* (GRPO) DeepSeek-AI et al. (2025). For every task–scene pair in a mini-batch we draw $G$ complete action sequences $\{\pi_{j,i}\}_{i=1}^{G}$ from the current policy, execute them once, and record the episode-level rewards $r_{j,i}$ defined in Section 4.3. **Group-Normalized Advantage.**    Within each group $j$ we compute the mean $\bar{r}_j$ and standard deviation $\sigma_j$, then form the advantage $A_{j,i} = \frac{r_{j,i} - \bar{r}_j}{\sigma_j}$.

**GRPO Surrogate Objective.**    Let $\rho_{j,i} = \pi_\theta(\pi_{j,i})/\pi_{\theta_{\text{old}}}(\pi_{j,i})$. The policy is updated by maximising

$$\mathcal{L}_{\text{GRPO}} = \frac{1}{BG} \sum_{j=1}^{B} \sum_{i=1}^{G} \min\big(\rho_{j,i} A_{j,i}, \text{clip}(\rho_{j,i}, 1-\varepsilon, 1+\varepsilon)\, A_{j,i}\big) - \beta\, D_{\text{KL}}\big[\pi_\theta \,\|\, \pi_{\text{ref}}\big],$$

where $\varepsilon$ is the clipping parameter and $\beta$ is the KL weight with respect to the reference model $\pi_{\text{ref}}$.

**Rewards coefficients**    In our experiments, the agent receives a penalty of -1 if its output action sequence cannot be parsed into a structured format. A base reward of 0.5 is granted for a successfully parsed sequence. The Semantic Relevance reward is scaled by a coefficient $\beta = 0.2$ and is capped at 1. The primary component is the Goal-Oriented Success reward, where the total reward for a task is 30, and the scaling factor $\alpha$ is defined as $30/N_{\text{sub}}$, with $N_{\text{sub}}$ being the number of sub-tasks.

## D    TECHNIQUE APPENDICES

Table 8 enumerates the skill primitives that the agent can compose into high-level programs. Each action takes a structured set of parameters, such as an object index, spatial relation, or target room, allowing the execution engine to translate symbolic action sequences into concrete state transitions. Together, these 13 primitives support navigation, object manipulation, state changes (e.g., open/close, toggle), and basic cooking operations, providing sufficient expressiveness for the long-horizon tasks used in our experiments.

## E    COMPARATION OF EMBODIED DATA GENERATION

We compare our Multi-Agent-Driven Automated Data Factory with other embodied scene generation works, as shown in Tab 4.

Table 5: Examples of task generation with/without social simulation in Scene Beechwood_0_garden

| With Social Simulation | Without Social Simulation |
|---|---|
| Pick up the basketballs in gym_0 and place them in the equipment rack in locker_room_0. | Please pick up the basketball from the storage rack in the gym_0 and place it on the mat in the gym_0. |
| Bring the yoga mats from locker_room_0 to gym_0 for Ms. Clara's yoga class. | Pick up the water bottle from the bench in locker_room_0 and place it inside the cabinet in locker_room_0. |
| Pick up the smoothie blender from corridor_0 and place it on the table in locker_room_0 for Ms. Clara's smoothie workshop. | Go to bathroom_0, toggle on the hand dryer, then toggle off the hand dryer after 20 seconds. |
| Gather sports bottles from corridor_1 and corridor_2 and place them in gym_0 for Coach Alex's basketball practice. | Pick up the volleyball from the storage box in locker_room_1 and place it on the counter in corridor_2. |
| Collect towels from locker_room_1 and distribute them in bathroom_0 for students to use after practice. | Open the locker in locker_room_1, pick up the towel inside, and place it on top of the bench in locker_room_1. |
| Pick up cleaning supplies from bathroom_0 and clean the floors in corridor_0 and corridor_1. | Pick up the gym bag from the bench in locker_room_0 and place it inside the storage cabinet in corridor_1. |
| Place the cones from corridor_2 in gym_0 for Coach Alex's basketball drills. | Go to gym_0, pick up the whistle from the referee table, and place it on the counter in corridor_0. |
| Toggle on the fans in gym_0 to ensure ventilation during Coach Alex's practice session. | Toggle on the light switch in corridor_2, then toggle off the light switch after 10 minutes. |
| Put the foam rollers from locker_room_1 inside locker_room_0 for Ms. Clara's wellness activities. | Pick up the cleaning spray from the shelf in corridor_0 and place it on top of the counter in bathroom_0. |
| Close the windows in corridor_0 to maintain temperature during Ms. Clara's fitness workshop. | Pick up the shoes from the floor in locker_room_1 and place them inside the shoe cabinet in locker_room_1. |

Table 6: **Prompt for the gpt4-based task diversity evaluator.**

Please analyze the diversity of the following generated commands and provide a score out of 10 based on the following criteria:
1. **Variety**: Do the commands include a diverse range of actions, objects, and scenarios? Do the commands cover different types of actions (e.g., picking up, placing, toggling) and involve various objects and locations? However the robot can only take concrete actions, such as pick up, move, toggle, place and so on, so don't be too strict about the action diversity.
2. **Inclusion of Specific Characters**: Do the commands explicitly mention specific individuals (their name or roles)? The characters number is limited to 2-4. Are the commands tailored to these characters, and do they reflect their unique roles or characteristics? If the commands don't include specific characters, please give a low score.
You need to be careful, just focus on the given criteria. After analyzing the commands, give your section scores and an overall score, provide a detailed explanation for the score.
Here are the commands: {commands}

Table 7: **Prompt for the gpt4-based scene aesthetic evaluator.**

You are a professional scene arrangement evaluator, capable of providing objective assessments of the reasonableness and aesthetics of each scene. Now, to complete a task, the user needs to place some new objects in an initial room, and these objects must satisfy certain spatial relationships, such as "A inside B" meaning B must be placed inside A, and so on. In the context of this task, please act as an evaluator to assess how well the user has arranged these new objects. We will provide you with an image, which is a top-down view of a room. The image will label the names of some objects, which are either newly added objects or initial objects that have spatial relationships with the newly added ones. Unlabeled objects are part of the room's original arrangement. Additionally, we will provide a JSON description of the new added objects that must be placed in this room, along with their spatial relationships that must be satisfied.
Here are some rules you must follow:
Step 1: Start with a full score of 10.
then:
1. Check Label Correspondence (Deduct 0–2 points)
- Verify whether the bounding boxes in the image match the objects specified in the JSON file.
- If there are mismatches, omissions, or incorrect names, deduct points accordingly:
- Minor mismatches (1–2 objects incorrect): Deduct 1 point
- Major mismatches (multiple objects incorrect or serious relational errors): Deduct 2 points
2. Assess Room Clutter (Deduct 0–4 points)
- Observe whether the room looks cluttered, whether objects are overlapping, crowded, or arranged chaotically.
- Deduct points as follows:
- Generally tidy, only slightly crowded: Deduct 1 point
- Noticeable crowding or some overlap: Deduct 2 points
- Multiple overlaps or moderate chaos but still recognizable: Deduct 3 points
- Extremely cluttered or unrecognizable: Deduct 4 points
3. Evaluate Aesthetics and Placement Reasonableness (Deduct 0–4 points)
- Consider whether objects are oriented naturally, placed reasonably, and visually harmonious.
- Deduct points as follows:
- Mostly reasonable with minor visual inconsistencies: Deduct 1 point
- Some objects have unnatural orientation or awkward positions: Deduct 2 points
- Several unreasonable placements or orientations: Deduct 3 points
- Most objects poorly placed or visually chaotic: Deduct 4 points
**Object Placements and Relationships**: {**new_added_tree**}
Please provide the score and a short explanation.

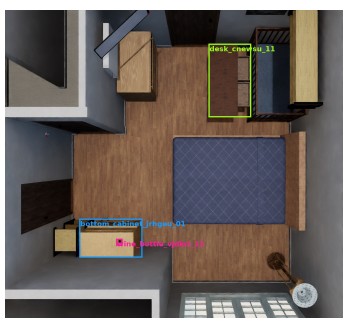
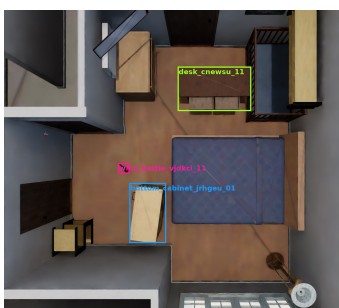
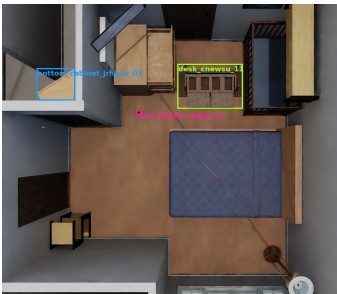

(a) Multi-level layout tree       (b) No multi-level layout tree       (c) LayoutGPT

Figure 9: Comparison of three methods when generating scene of Benevolence_2_int.

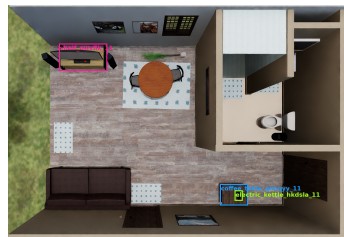
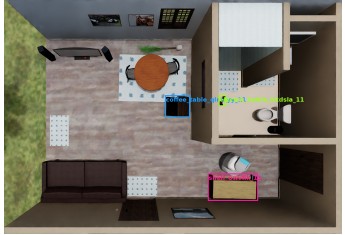
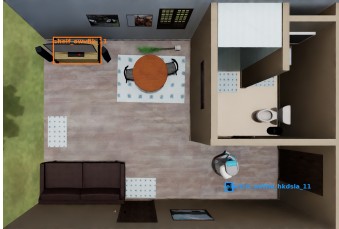

| (a) Multi-level layout tree | (b) No multi-level layout tree | (c) LayoutGPT |
|:---:|:---:|:---:|

Figure 10: Comparation of three methods when generating scene of Merom_0_garden.

Table 8: List of available action primitives and their parameters

| Action | Parameters |
|---|---|
| move | {object_index: $n$} |
| turn | {yaw: $y$} |
| pick_up | {object_index: $n$} |
| place | {object_index: $n$, relation: $r$} |
| move_forward | {distance: $x$, yaw: $y$} |
| open | {object_index: $n$} |
| close | {object_index: $n$} |
| toggle_on | {object_index: $n$} |
| toggle_off | {object_index: $n$} |
| heat_object_with_source | {object_index: $n$, source_index: $m$} |
| cook_object_with_tool | {object_index: $n$, source_index: $m$} |
| froze_object_with_source | {object_index: $n$, source_index: $m$} |
| go_to_room | {room_name: $s$} |

## F EXPERIMENTS DETAILS

We design two LLM-based evaluator for evaluating the generated tasks' diversity and the aesthetic score of the generated scenes. The two evaluator's prompts are shown in Tab. 6 and Tab. 7.

There are another two examples of three methods when generating scenes in Fig 9 and 10. We can see that the other two methods – layoutgpt and no layout tree give a messy and infeasible scene.

## G LLM USAGE STATEMENT

Large Language Models (LLMs) were used only as auxiliary tools in this work. Specifically, we employed GPT-based models to assist with grammar polishing, wording refinement, and summarization of related works. All core research components—including research ideation, methodology design, experiment execution, and result analysis—were conducted entirely by the authors. No LLM contributed at a level that would merit authorship.

