# OpenReview forum: "EmboMatrix: A Scalable Training-Ground for Embodied Decision-Making"
_ICLR.cc/2026/Conference — Submitted to ICLR 2026_

### Official Review · Reviewer_8Z5X · 2025-10-26

**Soundness:** 2
**Presentation:** 1
**Contribution:** 3
**Rating:** 4
**Confidence:** 4

**Summary:**

To bridge the gap between Large Language Models' (LLMs) reasoning abilities and the physical understanding required for embodied decision-making, this paper introduces EmboMatrix, a scalable "training ground" for interactive, simulation-based learning. EmboMatrix features three key innovations: a multi-agent data factory for generating massive and diverse tasks, a distributed heterogeneous-hardware system for scalable simulation, and a multi-level reward architecture for precise supervision. By training an LLM within this framework to create "EmboBrain," the resulting 7B model significantly outperformed much larger baselines, including the 671B DeepSeek-R1, on two challenging embodied decision-making benchmarks.

**Strengths:**

+ This work introduces EmboMatrix, the first "training ground" framework specifically proposed to enhance the embodied decision-making capabilities of LLMs through interactive learning.
+ The framework's effectiveness is clearly demonstrated by significant performance improvements; models trained within EmboMatrix (EmboBrain) substantially outperform much larger models, including the 671B DeepSeek-R1, on challenging benchmarks.
+ The system's architecturally decoupled design, which separates the LLM trainer from a distributed pool of heterogeneous simulation workers, enables high-throughput parallel rollouts and significantly improves the efficiency of the entire training loop.
+ The paper effectively uses an ablation study and reward curves to prove the necessity of its hierarchical reward design, showing that the 'semantic relevance reward' ($r_r$) is critical for guiding the agent and enabling efficient, stable learning.

**Weaknesses:**

+ The pre-cached language-physics interface, which substitutes full physical simulation with pre-computed post-conditions, sacrifices dynamic fidelity and generalization for speed. This approximation may lead to policies that lack robustness in the real world, as it omits crucial details like contact dynamics, sequential dependencies, and realistic failure modes. The paper does not provide a systematic error analysis of this approximation or address potential inconsistencies from pre-rendering.
+ The model, though the main focus is built on top of LLMs, with its inputs consisting of simulator-generated textual descriptions of the scene, not raw sensory data like visual perception, which bypasses the critical end-to-end challenge of perception and action.
+ Key details regarding the pre-caching system are not disclosed, including the storage overhead of the pre-computed outcomes, the runtime cache hit rate, and the fallback mechanism used when an interaction is not in the cache (e.g., whether it reverts to a full, costly physical simulation).
+ The statistical rigor of the evaluation is limited. The main results table only reports the average of 10 samples per task, without reporting variance, confidence intervals, or results from multiple independent random seeds, and lacks statistical significance testing.
+ The paper's formatting appears to deviate from the official ICLR 2026 template, seemingly using `\usepackage{geometry}` to alter the page layout.
+ There are minor formatting and typographical errors. For example, the caption in Figure 5 overlaps with text in the image, and some figure captions (Figures 2, 3) lack sufficient detail. There are also typos, such as "A detailed comparation is shown in Tab 4" (line 126) instead of "comparison", and "EmboMatrix improve its performance by" (line 337).

**Questions:**

1. How does the pre-caching approximation, which substitutes physics with post-conditions, affect the policy's robustness to real-world dynamics and unscripted failure modes?

2. How does the multi-agent data factory ensure genuine task diversity, and isn't this diversity ultimately bottlenecked by the limited set of objects and 45 base scenes from BEHAVIOR-1K?

3. What are the technical specifications of the pre-caching system, such as its storage overhead, typical runtime cache hit rate, and the fallback mechanism used for a cache miss?

4. To improve statistical rigor, can the authors provide variance, confidence intervals, or results run with multiple independent random seeds for the main benchmark results in Table 1?

---

> ### Author Response · Authors · 2025-11-25
> **Response Reviewer 8Z5X Part 1.**
>
> # Response to Weaknesses 1/ Question 1 by Reviewer 8Z5X:
> > W1: The pre-cached language-physics interface, which substitutes full physical simulation with pre-computed post-conditions, sacrifices dynamic fidelity and generalization for speed. This approximation may lead to policies that lack robustness in the real world, as it omits crucial details like contact dynamics, sequential dependencies, and realistic failure modes. The paper does not provide a systematic error analysis of this approximation or address potential inconsistencies from pre-rendering.
>
> > Q1: How does the pre-caching approximation, which substitutes physics with post-conditions, affect the policy's robustness to real-world dynamics and unscripted failure modes?
>
> We thank the reviewer for this insightful question. We fully agree that aspects such as contact dynamics, sequential dependencies, and realistic failure modes are central to Embodied AI. However, our adoption of the pre-cached language–physics interface is a deliberate strategic design choice. Our work is intended to be complementary to low-level control research, focusing specifically on solving the "decision-making efficiency" bottleneck rather than the "low-level control fidelity" problem.
> 1. Motivation: The Inevitable Trade-off for Reasoning Efficiency When training embodied models, a vast majority of computation is typically consumed by simulating low-level trajectories and calculating contact physics. This creates a conflict: to achieve high sample efficiency for high-level reasoning and planning, one must inevitably make trade-offs regarding the fidelity of low-level simulation.
>   1. Decoupling Reasoning from Control: Our goal is to explicitly decouple high-level planning from low-level execution. By abstracting away execution details (such as contact information and continuous dynamics) via pre-computed post-conditions, we allow the RL agent to focus its capacity entirely on long-horizon decision-making logic.
>   2. Assumption of Reliable Skills: This design operates under the standard hierarchical assumption that low-level skills are (or will be) reliable. This allows our method to serve as the "planner" that can be plugged into robust "controllers" developed by the control community.
> 2. Alignment with Recent Literature This decoupling strategy is consistent with a wide range of recent high-impact works. Our setting directly follows the Embodied Agent Interface (EAI) [1] and Unleashing Embodied Task Planning [2], which establish that high-level agents perform best when relieved of low-level dynamic computations. This paradigm is further validated by:
> - CoPa (CVPR 2024) [4]: Adopts a coarse-to-fine strategy where the foundation model identifies spatial constraints but explicitly abstracts away contact dynamics, delegating them to low-level solvers.
> - SayPlan (CoRL 2024) [5]: Argues that scalable long-horizon planning requires operating on abstract semantic representations rather than raw physical states to ensure planning feasibility.
>
> Thus, our "pre-cached" interface is not merely an approximation, but a standard justified design pattern aimed at solving the reasoning bottleneck. We acknowledge that this abstraction omits specific failure modes. In the final version, we will explicitly clarify this scope, positioning our framework as complementary to robust low-level control methods. We believe the integration of our high-level planner with advanced dynamics-aware controllers is a promising direction for future work.
>
> [1] Embodied Agent Interface: Benchmarking LLMs for Embodied Decision Making, NeurIPS 2025.
>
> [2] Unleashing Embodied Task Planning Ability in LLMs via Reinforcement Learning, arXiv 2024.
>
> [3] CoPa: General Robotic Manipulation through Spatial Constraints of Parts with Foundation Models, CVPR 2024.
>
> [4] SayPlan: Grounding Large Language Models using 3D Scene Graphs for Scalable Robot Task Planning, CoRL 2024.

---

> ### Author Response · Authors · 2025-11-25
> **Response Reviewer 8Z5X Part 2.**
>
> # Response to Weaknesses 2 by Reviewer 8Z5X:
> > The model, though the main focus is built on top of LLMs, with its inputs consisting of simulator-generated textual descriptions of the scene, not raw sensory data like visual perception, which bypasses the critical end-to-end challenge of perception and action.
>
> We thank the reviewer for raising this question. In our main experiments, we intentionally follow the exact setting of the Embodied Agent Interface (EAI), which is a pure-language benchmark. Because EAI defines the task interface, action sequencing protocol, and evaluation entirely in language form, we adopt the same assumptions to ensure comparability and to study high-level reasoning under the standard EAI formulation.
> To address your concern, we additionally constructed our own benchmark in which we incorporate vision-based scene-graph construction, following approaches such as TIGeR: Tool-Integrated Geometric Reasoning in Vision-Language Models for Robotics[1]. In this variant, we use visual tool calls to construct the scene graph required for task reasoning, thereby simulating a more realistic multimodal setting. Compared with the EAI setting—where the full scene graph is provided—the performance exhibits only a minor degradation, and the overall trends and core conclusions of our work remain unchanged.
> We appreciate your point regarding the integration of multimodal models. We agree that extending our framework to richer perceptual settings is a valuable direction, and we will highlight this as an important avenue for future work in the final version.
>
> **Table: Performance comparison on Our Agent-Generated Benchmark.**
>
> | Model | Overall | Pick and Place | Appliances Using | Kitchen Operation | Compound Task |
> | :--- | :---: | :---: | :---: | :---: | :---: |
> | DeepSeek-R1-Distillated Qwen2.5-7B | 5.5 | 11.9 | 5.6 | 2.7 | 14.6 |
> | EmboBrain-7B | 63.0 ± 3.1 | 69.4 ± 5.1 | 61.8 ± 4.8 | 57.5 ± 4.7 | 66.8 ± 3.8 |
> | EmboBrain-7B+DinoV2 | 57.3 ± 3.4 | 58.2 ± 5.3 | 54.0 ± 5.1 | 52.6 ± 3.0 | 66.1 ± 3.6 |
>
> [1] Han, Yi, et al. "TIGeR: Tool-Integrated Geometric Reasoning in Vision-Language Models for Robotics."
>
> # Response to Weaknesses 3/Question 3 by Reviewer 8Z5X:
> >Key details regarding the pre-caching system are not disclosed, including the storage overhead of the pre-computed outcomes, the runtime cache hit rate, and the fallback mechanism used when an interaction is not in the cache (e.g., whether it reverts to a full, costly physical simulation).
>
> We appreciate the reviewer's request for technical transparency regarding the pre-caching system. We provide the specific implementation details and metrics below:
> 1. System Construction & Fidelity: To ensure the fidelity of our abstraction, we employ an offline exhaustive sampling strategy. Using the APIs provided by IsaacSim and Behavior, we enumerate all plausible physical states of task-relevant objects and verify their transition dynamics.
> - Validation: We strictly verify that all initial and goal states defined in the BDDL files are physically reachable. Tasks failing this feasibility check are excluded.
> - Guarantee: For every task, we cache all valid predicates over relevant object pairs. This guarantees that if a skill's preconditions are met in the cache, the retrieved outcome reflects a 100% physically valid result (e.g., a valid placement pose is only assigned if Place(obj1, obj2) is physically feasible in the simulator).
> 2. Runtime Metrics (Hit Rate & Storage):
> - Cache Hit Rate: During our rigorous development phase, we performed multiple iterations of "failure discovery →cache rule refinement." In our final evaluation, the system achieved a cache hit rate exceeding 99.9%.
> 3. Fallback Mechanism: Regarding the fallback strategy: We do not revert to costly full physical simulation at runtime. Instead, to maintain high inference speed, a cache miss is treated as an execution failure (the plan is deemed infeasible). The model receives rewards only for formatting and semantic relevance, but zero for execution.
> - Rationale: Given our >99.9% hit rate, a cache miss almost rigorously implies a physically impermissible state within the defined world dynamics. Therefore, treating it as a failure is a "safety-first" design choice that prevents the agent from hallucinating physically impossible transitions.

---

> ### Author Response · Authors · 2025-11-25
> **Response Reviewer 8Z5X Part 3.**
>
> Response to Weaknesses 4/Question 4 by Reviewer 8Z5X:
> > The statistical rigor of the evaluation is limited. The main results table only reports the average of 10 samples per task, without reporting variance, confidence intervals, or results from multiple independent random seeds, and lacks statistical significance testing.
>
> To improve statistical rigor, can the authors provide variance, confidence intervals, or results run with multiple independent random seeds for the main benchmark results in Table 1?
> We thank the reviewer for identifying this issue. Over the past week, we have augmented our experimental analysis with confidence interval reporting (results averaged across 3 random runs). Updated results are presented in the table below. Going forward, we will incorporate additional random runs in the final version to further ensure statistical reliability.
> | Model | (Our) Overall | (Our) Pick and Place | (Our) Appliances Using | (Our) Kitchen Operation | (Our) Compound Task | (EAI) Overall | (EAI) Pick and Place | (EAI) Appliances Using | (EAI) Kitchen Operation | (EAI) Compound Task |
> | :--- | :---: | :---: | :---: | :---: | :---: | :---: | :---: | :---: | :---: | :---: |
> | 7B-Base | 5.5 | 11.9 | 5.6 | 2.7 | 14.6 | 4.1 | 1.0 | 4.9 | 1.3 | 6.8 |
> | EmboBrain-7B | 63.0 ± 3.1 | 69.4 ± 5.1 | 61.8 ± 4.8 | 57.5 ± 4.7 | 66.8 ± 3.8 | 60.0 ± 2.6 | 72.6 ± 3.0 | 66.1 ± 2.7 | 35.3 ± 2.5 | 58.6 ± 2.5 |
> | LLaMA3-8B | 5.3 | 11.1 | 3.7 | 0 | 9.7 | 3.3 | 1.2 | 2.5 | 2.2 | 5.3 |
> | LLaMA3-8B Enhanced | 57.9 ± 2.9 | 61.7 ± 6.3 | 57.9 ± 4.1 | 54.3 ± 3.5 | 60.1 ± 4.7 | 57.6 ± 2.3 | 70.0 ± 4.0 | 60.2 ± 3.9 | 33.1 ± 2.9 | 57.2 ± 2.4 |
>
> # Response to Weaknesses 5/6 by Reviewer 8Z5X:
> >5. The paper's formatting appears to deviate from the official ICLR 2026 template, seemingly using \usepackage{geometry} to alter the page layout.
> >6. There are minor formatting and typographical errors. For example, the caption in Figure 5 overlaps with text in the image, and some figure captions (Figures 2, 3) lack sufficient detail. There are also typos, such as "A detailed comparation is shown in Tab 4" (line 126) instead of "comparison", and "EmboMatrix improve its performance by" (line 337).
>
> Thanks for pointing it out! We will check these typos and format issues in the subsequent version.
>
> # Response to Question 2 by Reviewer 8Z5X:
> > How does the multi-agent data factory ensure genuine task diversity, and isn't this diversity ultimately bottlenecked by the limited set of objects and 45 base scenes from BEHAVIOR-1K?
>
> We thank the reviewer for this thoughtful comment regarding data diversity. We clarify that our multi-agent data factory ensures genuine diversity through combinatorial complexity and architectural extensibility, rather than being limited by the static count of base scenes.
>
> 1. Substantial Asset Scale Even within the current setup, the scale is extensive. We used BEHAVIOR-1K provided 7,722 objectsand 45 scenes spanning all major indoor categories (homes, hotels, offices, schools, gyms).
>
> 2. Combinatorial Task: Diversity Task diversity in our setting extends far beyond "one object in one scene." We leverage social simulation to generate tasks that vary along multiple orthogonal dimensions, creating a task space that is effectively exponential:
> - Temporal Complexity: Tasks range from short atomic actions to long-horizon procedures (e.g., 10 vs. 20+ steps).
> - Semantic Difficulty: Goals vary from simple interactions (heating a sandwich) to complex logic (preparing a full meal with dependency constraints).
> - Social Context: Uniquely, our factory introduces multi-agent dynamics, such as resolving conflicting requests or collaborative resource sharing, which adds a layer of variation unseen in single-agent benchmarks.
>
> Combine  substantial asset scale and above diversity, we generate more than 200k different tasks for agents to arrange suitable scene layout.
>
> 3. Open-Ended Architecture Crucially, our framework is not intrinsically bottlenecked by BEHAVIOR-1K. Built upon the Isaac Sim ecosystem, our pipeline supports loading any 3D assets in standard formats (e.g., USD, URDF). This means the system can readily incorporate new environments and objects from the broader graphics and robotics community(like omniverse assets library), ensuring that diversity is scalable and future-proof.
>
> # Summary
> We sincerely thank the reviewer for the suggestions regarding the integration with low-level physics to enhance completeness. We fully agree that the effective coordination between high-level planning and low-level execution is a pivotal direction for the future of embodied AI. Meanwhile, we maintain that these two levels possess distinct characteristics, each meriting dedicated training resources to maximize their respective capabilities. Finally, we are grateful for the constructive comments on visual inputs and confidence intervals, which have significantly helped us make this work more rigorous and solid. Thank you for your time and guidance!

---

### Official Review · Reviewer_SzXd · 2025-10-31

**Soundness:** 1
**Presentation:** 2
**Contribution:** 2
**Rating:** 2
**Confidence:** 3

**Summary:**

This paper introduce the EmboMatrix infrastructure and "training ground" for high-level embodied action sequencing using large language models (LLMs). EmboMatrix combines a multi-agent data factory, a distributed simulation backend , and a hierarchical reward architecture. The framework is used to train EmboBrain, an LLM adapted via reinforcement learning within simulated embodied environments.

**Strengths:**

1. The idea of scaling embodied environments to create a data engine for LLM training is interesting and somewhat novel. If executed well, can make a great resource for training embodied planners.

**Weaknesses:**

1. Authors claim strong performance gains on the EAI benchmark in abstract L26. However, experimental results on embodied agent interface that the authors report (Overall, Pick and Place, Appliances Using, Kitchen Operation, Compound Task) are not the standard axis of evaluation for EAI (Goal Interpretation, Subgoal Decomposition, Action Sequencing, Transition Modeling). The authors fail to clarify exactly what subset of tasks and what number of examples are used for each track, and the "Overall" score is highly misleading. It seems that the paper is only working on the Action Sequencing subtrack, which should be properly stated in obvious locations.

2. If the trained model is only capable of action sequence generation and the training envionment is only capable of generation action sequence prediction task data, then it is an overstatement to claim that the environment and model is for embodied decision making.

3. There's not enough comparison of EmboMatrix to other existing benchmarks and data generation methods, also there isn't sufficient analysis / experiments with "Our Agent-Generated Benchmark" to demonstrate it being a reliable benchmark.

**Questions:**

1. Typos: we present a comprehensive experiments (L264)
2. Additional citations: RoboVerse, Embodied World Models, etc.
3. How many tasks are there in each of the EAI eval subtracks (Pick and Place, Appliances Using, Kitchen Operation, Compound Task)?

---

> ### Author Response · Authors · 2025-11-25
> **Response Reviewer SzXd Part 1.**
>
> # Response to Weaknesses 1/2 by Reviewer SzXd:
> >1. Authors claim strong performance gains on the EAI benchmark in abstract L26. However, experimental results on embodied agent interface that the authors report (Overall, Pick and Place, Appliances Using, Kitchen Operation, Compound Task) are not the standard axis of evaluation for EAI (Goal Interpretation, Subgoal Decomposition, Action Sequencing, Transition Modeling). The authors fail to clarify exactly what subset of tasks and what number of examples are used for each track, and the "Overall" score is highly misleading. It seems that the paper is only working on the Action Sequencing subtrack, which should be properly stated in obvious locations.
> 2. If the trained model is only capable of action sequence generation and the training envionment is only capable of generation action sequence prediction task data, then it is an overstatement to claim that the environment and model is for embodied decision making.
>
> We thank the reviewer for scrutinizing our evaluation metrics and pointing out the ambiguity regarding the EAI benchmark tasks.
> We fully accept this criticism.  We modified the experiment setting decription in the updated version of PDF in line 337-line 338 that emphsize “Specifically, we focus on the Action Sequencing track in EAI benchmark."
>
> Rationale for Scope: While we acknowledge the presentation oversight, we respectfully maintain that this focus is scientifically justified for two key reasons:
> 1. Relevance to Interaction: The "Action Sequencing" track is the only component of EAI that necessitates dynamic environment interaction. Since our paper’s core contribution lies in enhancing LLM decision-making through active feedback and world modeling, this track provides the most direct signal for verifying our method. Other tracks (e.g., Goal Interpretation) focus more on static semantic parsing, which is less relevant to the interactive dynamics we aim to solve.
> 2. Definition of Embodied Decision Making: We posit that the fundamental essence of embodied decision-making is the generation of executable action sequences. This definition is not arbitrary but is well-supported by recent top-tier literature, which treats action generation as the primary metric for assessing decision-making capabilities:
>   - NVIDIA's Cosmos-Reason1 [1] explicitly states its goal is to "generate appropriate embodied decisions (e.g., next step action)" to validate physical understanding in its abstract part. Crucially, the paper evaluates the model's decision-making competence by directly measuring the accuracy of these generated next-step actions.
>   - RoboBrain (CVPR 2025) [2] places this capability at the core of its innovation. In its first contribution, the paper defines the essential challenge as "transforming abstract instructions into concrete actions." Accordingly, they evaluate their model specifically on planning tasks.
>   This choice of contribution and evaluation across top-tier works underscores a community consensus: planning (i.e., generating correct action sequences) is the most critical decision-making function of an embodied brain/embodied high-level model.
>
>
> [1] Azzolini et al., "Cosmos-Reason1: From Physical Common Sense to Embodied Reasoning", arXiv preprint, 2025.
>
> [2] Zhang et al., "RoboBrain: A Unified Brain Model for Robotic Manipulation from Abstract to Concrete", CVPR 2025.

---

> ### Author Response · Authors · 2025-11-25
> **Response Reviewer SzXd Part 2.**
>
> # Response to Weaknesses 3 by Reviewer SzXd:
> > There's not enough comparison of EmboMatrix to other existing benchmarks and data generation methods, also there isn't sufficient analysis/experiments with "Our Agent-Generated Benchmark" to demonstrate it being a reliable benchmark.
>
> To clarify the distinct contributions of EmboMatrix in data generation, we provide a systematic comparison with existing methods (see Table). Existing approaches fall into four main categories, none of which simultaneously satisfy the need for scale, semantic richness, and physical-task coupling required for complex embodied AI training:
> 1. Manual Authoring (e.g., BEHAVIOR-1K, ALFRED) provides high-quality, coupled tasks but suffers from high cost and limited scale and diversity.
> 2. Procedural Generation (e.g., ProcTHOR, VIMA-Bench) is scalable but "rule-bound," resulting in scenes (S) that are decoupled from rich task semantics (I).
> 3. Visual/Layout Generation (e.g., WorldCraft, DiffuScene) focuses on visual fidelity but, as we note in our paper (line 148), "ignore[s] task constraints or object interactions."
> 4. LLM-Driven (Offline Dataset): This is the most related category, but it has critical gaps. While RoboGen is limited to "small, simple layouts," other recent works like RoboGPT and AgentSense are critically non-scenic: they use LLMs to generate (Task, Plan) pairs or (Persona, Action) sequences but do not generate or instantiate the corresponding, physically executable 3D scenes (S) for these tasks.
> More importantly, recent research (i.e., "Mind the Gap", arXiv: 2508.00282) has empirically identified the fundamental flaw of relying on zero-shot LLMs for task generation: tasks generated by LLMs, while 'novel', are inherently "asocial" and "disembodied" .
> The Multi-Agent-Driven Automated Data Factory in EmboMatrix is designed to directly address this "generation gap" with a two-stage framework:
> 1. To solve the "asocial" problem: Instead of simple prompts, our "Multi-Agent Social Simulation" module (Appendix A.1) generates semantically rich, long-horizon instructions (I) and goals (G) that are grounded in a social context, providing inherent purpose.
> 2. To solve the "disembodied" problem: Instead of generating decoupled scenes, our "Multi-Level Hierarchical Scene Generation" module (Appendix A.2) takes the task's goals (G) and initial conditions (IC) as hard constraints. It then generates a fully task-aware and physically executable scene (S) through a coarse-to-fine, multi-level process.
>
> In summary, EmboMatrix is the first system to systematically couple socially-driven task semantics (I) with physically-grounded scene generation (S) at scale, providing the essential foundation for complex embodied decision-making.
>
> **Table: Comparison of Embodied Task and Scene Generation Methods**
> | Method Category | Representative Work(s) | Task Instruction (I) Generation | Scene (S) Generation | Task-Scene Coupling | Key Limitation(s) |
> | :--- | :--- | :--- | :--- | :--- | :--- |
> | 1. Manual Authoring | BEHAVIOR-1K, ALFRED, VirtualHome | Manually Designed | Manually Designed | **High** (but static) | Limited scale, diversity, and extremely high creation cost. |
> | 2. Procedural Generation | ProcTHOR, VIMA-Bench | Template-based | Rule-based Layouts | **Low** | "Rule-bound"; scalable but scenes are *decoupled* from rich task semantics. |
> | 3. Visual/Layout Generation | WorldCraft, DiffuScene, HOLODECK | N/A or Simple Prompt | LLM/Diffusion-aided Visuals | **Very Low** | Focuses on visual fidelity; *ignores task constraints* and physical interactivity. |
> | 4. LLM-Driven (Offline Dataset) | RoboGen | Single LLM | LLM-gen Simple Layouts | **Medium** | Generation is tied to small, simple layouts. |
> | | RoboGPT | LLM (Self-instruction) | **N/A** (Does not generate scene) | **Medium** (Task-Plan only) | Generates (Task, *Plan*) pairs, but **not** the corresponding 3D scene (S). |
> | | AgentSense | LLM (3-stage Persona) | **N/A** (Does not generate scene) | **Medium** (Task-Action only) | Generates (Persona, *Action*) sequences, but **not** the corresponding task scene (S). |
> | **5. EmboMatrix (Ours)** | **EmboMatrix** | **Multi-Agent Social Simulation** | **Multi-Level Hierarchical Generation** | **Very High (Systematic)** | **(Our Contribution)** Systematically addresses all above limitations. |
>
> # Response to Question 1 by Reviewer SzXd:
> >Typos: we present a comprehensive experiments (L264)
>
> Thanks for pointing it out! We will check this typo in the subsequent version. We corrected this typo in the updated pdf.

---

> ### Author Response · Authors · 2025-11-25
> **Response Reviewer SzXd Part 3.**
>
> # Response to Question 2 by Reviewer SzXd:
> > Additional citations: RoboVerse, Embodied World Models, etc.
>
> We thank the reviewer for highlighting these critical related works. We explicitly cite and discuss them to contextualize our contribution at line 126 and line 131 of the  updated pdf.
>
> On RoboVerse: We acknowledge RoboVerse [1] as a landmark contribution to the field. It introduces a unified, multi-simulator framework that standardizes physics-consistent data generation and robot embodiment modeling. By enabling robust low-level skill acquisition across diverse rendering and physics engines, RoboVerse provides the ideal "motor cortex" foundation. Our work complements this by focusing on the "prefrontal cortex"—leveraging these reliable low-level capabilities (abstracted via our interface) to solve long-horizon reasoning tasks.
>
> On Embodied World Models: Similarly, recent advances in Embodied World Models, such as DayDreamer [2] and UniSim [3], demonstrate the power of learning internal predictive models of physical dynamics (e.g., via latent dynamics or neural simulation) to improve sample efficiency.
>
> References:
> [1] Geng, H., et al. "RoboVerse: Towards a Unified Platform, Dataset and Benchmark for Scalable and Generalizable Robot Learning", RSS 2025.
> [2] Wu, P., et al. "DayDreamer: World Models for Physical Robot Learning", CoRL 2023.
> [3] Yang, Sherry, et al. "Learning Interactive Real-World Simulators." ICLR 2024.
>
> # Response to Question 3 by Reviewer SzXd:
> > How many tasks are there in each of the EAI eval subtracks (Pick and Place, Appliances Using, Kitchen Operation, Compound Task)?
>
> | BenchMark | Overall | Pick and Place | Appliances Using | Kitchen Operation | Compound Task |
> | :--- | :---: | :---: | :---: | :---: | :---: |
> | Our Agent-Generated Benchmark | 100 | 21 | 18 | 37 | 24 |
> | EAI Action Sequencing Benchmark | 100 | 28 | 13 | 15 | 44 |
>
> # Summary:
> We sincerely thank the reviewer for identifying the ambiguity in our description and apologize for any confusion caused. However, we respectfully maintain that action sequencing stands as a fundamental benchmark in embodied decision making. Given that this evaluation protocol is widely adopted across numerous related works, we hope the reviewer recognizes the significance of the substantial progress our method has achieved in this critical domain.

---

### Official Review · Reviewer_F41E · 2025-11-02

**Soundness:** 3
**Presentation:** 3
**Contribution:** 3
**Rating:** 6
**Confidence:** 3

**Summary:**

This paper proposes EmboMatrix, a training ground for embodied decision making. The system has three key components: a multi-agent data engine for scene synthesis, a decoupled simulator with pre-cached language-to-physics mappings, and a hierarchical reward curriculum. The curriculum progresses from format correctness to semantic relevance and goal completion. Models trained in EmboMatrix acquire robust planning abilities through interactive learning. The resulting system, EmboBrain, substantially outperforms competitive baselines.

**Strengths:**

1. The paper is well-written, with clear figures and detailed explanations that make the work easy to follow.
2. The paper tackles a central gap in embodied AI by proposing a unified training ground that connects language-only LLMs to physical, interactive decision-making.
3. The system is well-engineered, combining multi-agent task generation, a scalable decoupled simulator with a pre-cached language–physics interface, and a hierarchical reward curriculum.
4. The evaluations are compelling and scale-efficient. A smaller EmboBrain model consistently surpasses a much larger DeepSeek-R1 across multiple embodied decision-making benchmarks and task categories, underscoring the practical impact of the unified training ground.

**Weaknesses:**

1. The method relies on a fixed, predefined skill library; the high-level policy can only select from these primitives, so generalization is limited by their coverage and granularity, and the set is not learned or expanded online.

2. The evaluation relies on GPT-4 to score task diversity and scene aesthetics, which risks creating a self-referential loop. When an LLM-trained system is judged by another LLM, results may be biased toward the evaluator's preferences.

3. The paper has limited environmental coverage. The training corpus and evaluation are built on 45 Behavior-based scenes, leaving cross-domain generalization and robustness to distribution shift unclear.

**Questions:**

1. The experiments focus exclusively on the DeepSeek-distilled Qwen series. Including results from a different model family would help demonstrate the generalization more broadly.

---

> ### Author Response · Authors · 2025-11-25
> **Response Reviewer F41E Part 1.**
>
> # Response to Weaknesses 1 by Reviewer F41E:
> > The method relies on a fixed, predefined skill library; the high-level policy can only select from these primitives, so generalization is limited by their coverage and granularity, and the set is not learned or expanded online.
>
> We thank the reviewer for this insightful observation. We acknowledge that relying on a fixed skill library theoretically places an upper bound on the types of atomic actions an agent can perform, compared to fully learning skills from scratch.
> However, we respectfully argue that in practice, this design choice does not severely constrain generalization for the following reasons:
>
> 1. Combinatorial Generalization: The core power of our method lies in the temporal compositionality of these skills. Much like a finite alphabet can construct an infinite number of sentences, our fixed set of atomic primitives allows the high-level policy to sequence actions into a vast array of complex behaviors. This combinatorial flexibility ensures sufficient expressiveness: in our experiments, this exact skill library enabled the successful generation and execution of thousands of distinct tasks across diverse semantic contexts. This demonstrates that the bottleneck for generalization is rarely the granularity of the primitives, but rather the capability to orchestrate them effectively.
> 2. Alignment with SOTA Paradigms: Decomposing complex behaviors into robust atomic primitives is a well-established paradigm in hierarchical planning. The primary motivation for fixing the skill set is to abstract away low-level control noise and provide a stable action space for high-level reasoning. By treating skills as reliable functional units, we avoid the instability inherent in non-stationary low-level learning, allowing the high-level policy to focus effectively on long-horizon logic and sequencing. Recent top-tier research continues to validate the efficacy of this approach. For example, Odyssey (IJCAI 2025) [1] demonstrates that a fixed library of 40 atomic skills enables agents to perform open-ended exploration in massive environments. Similarly, GLIDER (ICML 2025) [2] employs fixed low-level controllers as a stable basis for learning high-level abstractions in long-horizon tasks.
>
> Therefore, our adoption of a standardized primitive set (following EAI, NeurIPS 2024) [3] is consistent with the current trajectory of scalable embodied AI research.
>
> [1] Liu, Shunyu, et al. "Odyssey: Empowering Minecraft Agents with Open-World Skills."
> Proceedings of the Thirty-Fourth International Joint Conference on Artificial Intelligence, IJCAI 2025
>
> [2] Hu, Zican, et al. "Divide and Conquer: Grounding LLMs as Efficient Decision-Making Agents via Offline Hierarchical Reinforcement Learning." Forty-second International Conference on Machine Learning, ICML 2025.
>
> [3] Li, Manling, et al. "Embodied agent interface: Benchmarking llms for embodied decision making." Advances in Neural Information Processing Systems 37 (2024), NeurIPS 2024.
>
> # Response to Weaknesses 2 by Reviewer F41E:
> > The evaluation relies on GPT-4 to score task diversity and scene aesthetics, which risks creating a self-referential loop. When an LLM-trained system is judged by another LLM, results may be biased toward the evaluator's preferences.
>
> Thank you for raising this critical point. We acknowledge the concern regarding potential bias when an LLM-trained system is evaluated by a similar model.
>
> To strictly validate our findings and eliminate potential self-referential bias, we expanded our evaluation to include human users and multiple independent LLMs (Grok-4, Gemini, and GPT-5). Specifically for human study, we invited 7 human evaluators to score both "Task Diversity" and "Scene Layout Rationality", adhering to the exact same scoring criteria used by the AI evaluators.
>
> As shown in Table 1 and Table 2 below, the results from human evaluators and other LLMs align consistently with our original GPT-4 evaluations. For instance, in Scene Aesthetics, our method achieves the highest scores in average across all evaluators, significantly outperforming the baselines. This cross-validation provides strong evidence that the performance gains are attributable to our proposed method rather than artifacts of a specific evaluator model.
>
> Table 1: Evaluation of Task Generation Diversity across different evaluators.
> | Method | Human | Grok-4 | Gemini-2.5 Pro | GPT-5 |
> | :--- | :--- | :--- | :--- | :--- |
> | With Social Simulation | 6.83 | 8.16 | 9.51 | 8.91 |
> | Without Social Simulation | 3.74 | 4.98 | 5.02 | 4.64 |
> | Ratio (social > without) | 98% | 100% | 98% | 100% |
>
> Table 2: Evaluation of Scene Aesthetics compared with baselines.
> | Method | Human | Grok-4 | Gemini-2.5 Pro | GPT-5 |
> | :--- | :--- | :--- | :--- | :--- |
> | Our | 8.69 | 7.39 | 6.90 | 7.62 |
> | No tree | 5.69 | 7.40 | 6.77 | 7.50 |
> | LayoutGPT | 2.74 | 5.85 | 5.83 | 7.20 |

---

> ### Author Response · Authors · 2025-11-25
> **Response Reviewer F41E Part 2.**
>
> # Response to Weaknesses 3 by Reviewer F41E:
> > The paper has limited environmental coverage. The training corpus and evaluation are built on 45 Behavior-based scenes, leaving cross-domain generalization and robustness to distribution shift unclear.
>
> We thank the reviewer for this thoughtful comment regarding data diversity. We clarify that our multi-agent data factory ensures genuine diversity through combinatorial complexity and architectural extensibility, rather than being limited by the static count of base scenes.
>
> 1. Substantial Asset Scale Even within the current setup, the scale is extensive. We used BEHAVIOR-1K provided 7,722 objectsand 45 scenes spanning all major indoor categories (homes, hotels, offices, schools, gyms).
>
> 2. Combinatorial Task: Diversity Task diversity in our setting extends far beyond "one object in one scene." We leverage social simulation to generate tasks that vary along multiple orthogonal dimensions, creating a task space that is effectively exponential:
> - Temporal Complexity: Tasks range from short atomic actions to long-horizon procedures (e.g., 10 vs. 20+ steps).
> - Semantic Difficulty: Goals vary from simple interactions (heating a sandwich) to complex logic (preparing a full meal with dependency constraints).
> - Social Context: Uniquely, our factory introduces multi-agent dynamics, such as resolving conflicting requests or collaborative resource sharing, which adds a layer of variation unseen in single-agent benchmarks.
>
> Combine  substantial asset scale and above diversity, we generate more than 200k different tasks for agents to arrange suitable scene layout.
>
> 3. Open-Ended Architecture Crucially, our framework is not intrinsically bottlenecked by BEHAVIOR-1K. Built upon the Isaac Sim ecosystem, our pipeline supports loading any 3D assets in standard formats (e.g., USD, URDF). This means the system can readily incorporate new environments and objects from the broader graphics and robotics community(like omniverse assets library), ensuring that diversity is scalable and future-proof.
>
> # Response to Question 1 by Reviewer F41E:
> > The experiments focus exclusively on the DeepSeek-distilled Qwen series. Including results from a different model family would help demonstrate the generalization more broadly.
>
> We appreciate the suggestion to verify the universality of our method. In response, we have conducted additional training on the LLaMA-3-8B model to test our approach on a completely different base architecture. As shown in the table below, applying our method to LLaMA-3-8B results in a massive performance leap (improving the overall score from 5.3% to 57.9%).
>
>
> | Model | (Our) Overall | (Our) Pick and Place | (Our) Appliances Using | (Our) Kitchen Operation | (Our) Compound Task | (EAI) Overall | (EAI) Pick and Place | (EAI) Appliances Using | (EAI) Kitchen Operation | (EAI) Compound Task |
> | :--- | :---: | :---: | :---: | :---: | :---: | :---: | :---: | :---: | :---: | :---: |
> | 7B-Base | 5.5 | 11.9 | 5.6 | 2.7 | 14.6 | 4.1 | 1.0 | 4.9 | 1.3 | 6.8 |
> | EmboBrain-7B | 63.0 ± 3.1 | 69.4 ± 5.1 | 61.8 ± 4.8 | 57.5 ± 4.7 | 66.8 ± 3.8 | 60.0 ± 2.6 | 72.6 ± 3.0 | 66.1 ± 2.7 | 35.3 ± 2.5 | 58.6 ± 2.5 |
> | LLaMA3-8B | 5.3 | 11.1 | 3.7 | 0 | 9.7 | 3.3 | 1.2 | 2.5 | 2.2 | 5.3 |
> | LLaMA3-8B Enhanced | 57.9 ± 2.9 | 61.7 ± 6.3 | 57.9 ± 4.1 | 54.3 ± 3.5 | 60.1 ± 4.7 | 57.6 ± 2.3 | 70.0 ± 4.0 | 60.2 ± 3.9 | 33.1 ± 2.9 | 57.2 ± 2.4 |
>
>
> # Summary
> We sincerely appreciate the reviewer's constructive feedback regarding the evaluation metrics for task generation and the testing of multiple base models. We agree that these additional experiments have significantly strengthened the completeness and robustness of our work. Thank you for your time and valuable suggestions!

---

### Official Review · Reviewer_LYJB · 2025-11-04

**Soundness:** 3
**Presentation:** 3
**Contribution:** 3
**Rating:** 6
**Confidence:** 4

**Summary:**

This paper proposes a 'training ground' for LLM-based embodied agents. This environment, referred to as 'EmboMatrix', provides an end-to-end framework that encompasses scene generation, simulation for model rollouts, and reward signals for both training and evaluation. The authors introduce various techniques to optimize the simulation and rollout processes.  LLM models were subsequently trained using the EmboMatrix. Experimental results indicate that the resulting model outperformed both the base models and general-purpose, large-parameter models when evaluated on scenes generated within the EmboMatrix.

**Strengths:**

1. The reviewer appreciates the framework's ability to enable high-throughput rollouts, noting that rollout speed is often a bottleneck in embodied AI RL training. The proposed components—specifically the pre-cached language-physics interface, resource scheduler, and task dispatcher—appear to work well. The ablation study also shows they improve simulation speed, which the reviewer agrees is critical for RL practitioners.

2. This paper is well-structured and very easy to read. While some further clarification is needed, most of the technical details are clearly presented.

3. The experimental results look promising, the EmboBrain models outperform baseline models by and large general purpose LLMs.

**Weaknesses:**

1. The current documentation for the multi-agent-driven automated data factory lacks essential details regarding the intricate interactions between its various components. Specifically, the reviewer requires a more comprehensive explanation of how these multiple agents communicate, coordinate, and influence the generation of instructions and subsequent scene constructions. Without this crucial clarification, the underlying mechanisms and overall impact on the data factory's output remain ambiguous and difficult to assess.

2. Confidence intervals are not included in the results. Given the potentially high variance of RL methods, the absence of confidence intervals makes it challenging to accurately assess and compare the different approaches.

**Questions:**

1. In the pre-cached language-physics interface, physically plausible outcomes from a pre-computed set are used instead of running a full simulation. This raises two critical questions: first, what is the accuracy of this approximation, and second, is this technique employed solely during training or also during evaluation?

2. In the context of EmboMatrix rollout and training, which aspect consumes the most time: the environment step, model inference, or gradient update? Please provide a detailed explanation.

3. Regarding the parallel rollouts, are the model updates performed synchronously or asynchronously? If the updates are synchronous, could you please comment on the associated synchronization overhead? Conversely, if the updates are asynchronous, what impact does this have on training performance?

4. Regarding the semantic reward, could you elaborate on the design rationale for basing it on the intersection of needed objects and objects the agent has interacted with? How does this reward structure generalize to tasks that are not object-centric? Additionally, did you experiment with alternative reward formulations?

5. Given that intermediate rewards are often prone to reward hacking, did you observe any such unintended behaviors or policy exploitation stemming from this specific design?

---

> ### Author Response · Authors · 2025-11-25
> **Response Reviewer LYJB Part 1.**
>
> # Response to Weaknesses 1 by Reviewer LYJB:
> > The current documentation for the multi-agent-driven automated data factory lacks essential details regarding the intricate interactions between its various components. Specifically, the reviewer requires a more comprehensive explanation of how these multiple agents communicate, coordinate, and influence the generation of instructions and subsequent scene constructions. Without this crucial clarification, the underlying mechanisms and overall impact on the data factory's output remain ambiguous and difficult to assess.
>
> To address the reviewer’s concern, we clarify that our automated data factory is driven by a pipeline of communicating agents whose outputs form the structured inputs of the subsequent modules, and whose downstream errors propagate upward for selective regeneration. Specifically, this agent coordination occurs in two main stages: instruction generation and scene generation.
> 1. In the instruction-generation stage, scene metadata first conditions a Role-Playing Agent, whose character profiles are passed to a Social Simulation Agent; its task narratives are then forwarded to a Summarization Agent to produce BDDL goals. Each agent operates on the explicit outputs of the previous one, ensuring that the social context, scene semantics, and final task definition remain consistent.
> 2. In the scene-generation stage, the BDDL goal triggers a hierarchical chain of four agents—scene, room, planar, and object levels. The Scene-level Distribution Agent assigns rooms for all task-relevant objects and provides this to the Room-level Organization Agent, which constructs a relation tree that guides the Planar-Level Placement Generator and ultimately the Object-level Sampling Agent. Communication here is both top–down (each level constrains finer-grained decisions) and bottom–up: if a lower-level placement fails due to collisions, infeasible relations, or unreachable poses, the error is returned to the responsible higher-level agent, which revises only the affected part.
>
> This bidirectional flow—sequential conditioning from above and corrective feedback from below—ensures that the agents not only coordinate but actively shape each other’s outputs. As a result, both the generated instructions and the constructed scenes remain coherent, executable, and physically valid.
>
> # Response to Weaknesses 2 by Reviewer LYJB:
> > Confidence intervals are not included in the results. Given the potentially high variance of RL methods, the absence of confidence intervals makes it challenging to accurately assess and compare the different approaches.
>
> We agree with the reviewer that reporting confidence intervals is essential for accurately assessing RL methods due to their inherent variance.
>
> In response to this valid concern, we have re-executed the experiments to include statistical reporting. The table below presents the updated results (mean ± standard deviation) averaged across 3 random seeds.
>
> Please note that due to the limited time window of the rebuttal phase, the table below specifically highlights the experiments added during this discussion period. We are currently running the remaining baselines and commit to including confidence intervals for all training models in the final camera-ready version to ensure comprehensive statistical reliability.
>
> | Model | (Our) Overall | (Our) Pick and Place | (Our) Appliances Using | (Our) Kitchen Operation | (Our) Compound Task | (EAI) Overall | (EAI) Pick and Place | (EAI) Appliances Using | (EAI) Kitchen Operation | (EAI) Compound Task |
> | :--- | :---: | :---: | :---: | :---: | :---: | :---: | :---: | :---: | :---: | :---: |
> | 7B-Base | 5.5 | 11.9 | 5.6 | 2.7 | 14.6 | 4.1 | 1.0 | 4.9 | 1.3 | 6.8 |
> | EmboBrain-7B | 63.0 ± 3.1 | 69.4 ± 5.1 | 61.8 ± 4.8 | 57.5 ± 4.7 | 66.8 ± 3.8 | 60.0 ± 2.6 | 72.6 ± 3.0 | 66.1 ± 2.7 | 35.3 ± 2.5 | 58.6 ± 2.5 |
> | LLaMA3-8B | 5.3 | 11.1 | 3.7 | 0 | 9.7 | 3.3 | 1.2 | 2.5 | 2.2 | 5.3 |
> | LLaMA3-8B Enhanced | 57.9 ± 2.9 | 61.7 ± 6.3 | 57.9 ± 4.1 | 54.3 ± 3.5 | 60.1 ± 4.7 | 57.6 ± 2.3 | 70.0 ± 4.0 | 60.2 ± 3.9 | 33.1 ± 2.9 | 57.2 ± 2.4 |

---

> ### Author Response · Authors · 2025-11-25
> **Response Reviewer LYJB Part 2.**
>
> # Response to Question 1 by Reviewer LYJB:
> > In the pre-cached language-physics interface, physically plausible outcomes from a pre-computed set are used instead of running a full simulation. This raises two critical questions: first, what is the accuracy of this approximation, and second, is this technique employed solely during training or also during evaluation?
>
> We observe that when training embodied models in a fully end-to-end manner, from high level to low level. The overwhelming majority of training compute and gradient updates are consumed by fitting low-level motor trajectories and other fine-grained control behaviors. This motivates our core objective: to decouple high-level planning from low-level manipulation during training. In this work, we focus exclusively on improving the capability of the high-level model.
> To achieve this, we introduce a pre-cached language–physics interface that provides a highly abstracted representation of low-level consequences, along with pre-sampled outcomes of low-level skill executions. This design allows the high-level policy to reason over semantically meaningful transitions without incurring the cost of simulating full-fidelity physical executions at training time.
>
> Regarding the two questions you raised:
>
> 1. The accuracy of this approximation.
> To ensure the fidelity of our abstraction, we exhaustively sample all plausible physical states of task-relevant objects using the APIs provided by IsaacSim and Behavior. These sampled transitions are aligned with the action-sequencing conventions defined in the Embodied Agent Interface (EAI) , NeurIPS 2024. During task and dataset construction, we additionally verify that all initial and goal states defined in the BDDL file are physically reachable; tasks that fail this feasibility check are excluded from both training and evaluation.
> Furthermore, for every task, we enumerate and cache all predicates over relevant object pairs as defined by the simulator. This guarantees that whenever the preconditions of a skill are satisfied, the corresponding outcome in our cache reflects a 100% physically valid result. For example, when invoking Place(object1, object2, "ontop"), if object1 is grasped and the robot is within manipulable distance of object2, and the cached predicate object1 ontop object2 is feasible, we directly assign object1 to a valid sampled placement pose.
> Crucially, this design enables fast and accurate rollout validation: the high-level policy always encounters outcomes consistent with EAI’s execution semantics, while avoiding costly low-level physics simulation. As a result, the training process remains focused on improving high-level decision making rather than repeatedly fitting low-level motor trajectories.
>
> 2. Consistency during training and evaluation.
> To ensure alignment with the Embodied Agent Interface, we apply the same abstraction and feasibility assumptions during both training and evaluation. This ensures that the high-level model is trained under the exact abstraction it will encounter at test time.
>
> # Response to Question 2 by Reviewer LYJB:
> > In the context of EmboMatrix rollout and training, which aspect consumes the most time: the environment step, model inference, or gradient update? Please provide a detailed explanation.
>
> After incorporating our full set of simulation–optimization techniques, the end-to-end training loop—comprising rollout sampling, simulator-based reward computation, and parameter updates—tends to split roughly in a 4:3:3 time ratio. Of course, this ratio is also highly dependent on the underlying infrastructure. In our current Kubernetes-based distributed simulation cluster, if all nodes are equipped with high-performance SSDs for storing simulation-related data, the entire simulation pipeline can be further accelerated.

---

> ### Author Response · Authors · 2025-11-25
> **Response Reviewer LYJB Part 3.**
>
> # Response to Question 3 by Reviewer LYJB:
> > Regarding the parallel rollouts, are the model updates performed synchronously or asynchronously? If the updates are synchronous, could you please comment on the associated synchronization overhead? Conversely, if the updates are asynchronous, what impact does this have on training performance?
>
> Our entire pipeline operates synchronously within the VERL reinforcement learning framework[1]. Concretely, for each training batch, we may sample—for example—8 tasks, each with 100 rollouts. Given that the simulation backend launches 2 simulators per task, the resulting 800 rollouts are distributed in parallel across all available simulators. Because our design aggressively bypasses expensive low-level physics simulation, the runtime variance across tasks is minimal. Empirically, the simulation component accounts for only ~3.5% of the total wall-clock time per training step on average, and never exceeds 10% in the worst case. This efficiency ensures that the majority of computation is devoted to optimizing high-level decision making rather than waiting on heterogeneous low-level simulation dynamics.
>
> [1]: Sheng, Guangming, et al. "Hybridflow: A flexible and efficient rlhf framework." Proceedings of the Twentieth European Conference on Computer Systems. 2025.
>
> # Response to Question 4 by Reviewer LYJB:
> > Regarding the semantic reward, could you elaborate on the design rationale for basing it on the intersection of needed objects and objects the agent has interacted with? How does this reward structure generalize to tasks that are not object-centric? Additionally, did you experiment with alternative reward formulations?
>
> In our initial experiments, we adopted the format reward and task-success reward described in the DeepSeek-R1 paper. However, for the 1.5B–7B models we train, zero-shot rollouts very rarely produce feasible solutions. This leads to extremely low GRPO efficiency. The semantic-intersection reward is therefore introduced primarily for two reasons:
> (1) to restrict the sampling space of smaller models, and
> (2) to leverage the task-generation framework, which directly provides access to the set of task-relevant objects.
> This significantly improves early-stage rollout feasibility and stabilizes training dynamics.
> Regarding generalization to non–object-centric tasks, we clarify that although the Behavior/EAI action-sequencing benchmark is primarily object-centric, our reward formulation is not limited to object nouns. Location entities (e.g., living room, shelf, sink area) are handled analogously and incorporated into the semantic reward whenever they appear in the BDDL specification. Thus, for BDDL-style programmatic task definitions, the semantic reward naturally applies to a broader range of semantic categories and is not restricted to physical objects.
> On alternative reward formulations, our current experience is that the proposed semantic reward is already minimal, unambiguous, and highly effective at improving sampling efficiency. We have not explored alternative reward functions in the present submission. Beyond reward design, we considered using a small-scale SFT “cold start” to help the model focus on higher-value sampling regions as a replacement for the semantic reward. However, because this procedure is not purely reinforcement-learning-driven and would require additional curated supervision, we chose not to include it in the main method.
> Finally, we added a comparison between the semantic reward and an SFT-based warm start under a fixed training budget of 50 steps. Due to time constraints, a full evaluation with confidence intervals will be included in the appendix of the final version.
>
> Table: 50-step RL performance comparison on Our Agent-Generated Benchmark.
>
> | Model | Overall | Pick and Place | Appliances Using | Kitchen Operation | Compound Task |
> | :--- | :---: | :---: | :---: | :---: | :---: |
> | EmboBrain-7B-cold start | 26.5 | 35.7 | 35.0 | 15.3 | 29.2 |
> | EmboBrain-7B-semantic reward | 25.8 | 28.3 | 29.4 | 17.1 | 28.3 |

---

> ### Author Response · Authors · 2025-11-25
> **Response Reviewer LYJB Part 4.**
>
> # Response to Question 5 by Reviewer LYJB:
> >Given that intermediate rewards are often prone to reward hacking, did you observe any such unintended behaviors or policy exploitation stemming from this specific design?
>
> Our use of the semantic reward is intentionally highly conservative. Its magnitude is extremely small and is capped by a strict upper bound—the maximum possible semantic reward is 30× smaller than a single task-success reward. The semantic reward is therefore not designed to drive long-term behavior; instead, its role is limited to the very early phase of training, where smaller models exhibit extremely low zero-shot feasibility and thus struggle to obtain any task-success signal. In this regime, the search space is excessively large, and the semantic reward serves only to gently bias the model toward task-relevant entities until it can reliably reach meaningful success states.
> After the model surpasses this initial phase, the semantic reward becomes negligible due to both its small scale and its capped contribution. Empirically, we have not observed any indication of reward hacking or pathological optimization behavior. The high-level policy quickly shifts to optimizing the dominant task-success reward once it becomes reachable, and the semantic reward plays no significant role beyond early-phase stabilization. This controlled design ensures that the reward shaping improves sampling efficiency without distorting the intended optimization objective.
>
> # Summary
> We sincerely thank you for taking the time to read and evaluate our work. Your review reflects a deep understanding of the area, and we truly appreciate the professionalism, fairness, and insightfulness of your feedback. It is increasingly rare in modern AI conferences to encounter reviewers who engage with this level of expertise and care. If you have any further questions or would like to discuss additional aspects of the work, we would be very happy to continue the conversation.

---

### Author Response · Authors · 2025-12-03
**Summary of discussions During Rebuttal Phase.**

Across all reviews, the main questions centered on:

1. The clarity and diversity of our multi-agent data factory.
2. The fidelity and role of the pre-cached language–physics interface.
3. RL training details and statistical rigor.
4. Generalization beyond a single backbone and beyond pure language input.
5. The precise scope of our EAI evaluation and overall claims.

Through the additional analyses and experiments in this rebuttal, we believe we have substantively addressed these concerns:

1. We clarified in detail how the multi-agent data factory communicates and coordinates (top–down conditioning and bottom–up error propagation), and showed that diversity comes from combinatorial social simulation plus extensible assets, not just the 45 BEHAVIOR-1K base scenes, yielding **200k+ executable tasks** and a pipeline that can ingest new USD/URDF assets.
2. We explained the design of the pre-cached language–physics interface as a deliberate abstraction for high-level reasoning: how it is constructed by exhaustive offline sampling, how it achieves >99.9% hit rate, how training and evaluation share the exact same abstraction, and how cache misses are handled conservatively as failures. We also explicitly repositioned it as complementary to low-level control and Sim2Real, not a replacement.
3. We detailed the RL training loop: fully synchronous rollouts in VERL/HybridFlow, the empirical time breakdown (≈4:3:3 across env, inference, and updates), and the small contribution of simulation to wall-clock time. For the semantic reward, we explained its motivation, conservative scaling, applicability to both objects and locations, and presented ablations (including comparison to SFT cold-start), together with multi-seed runs and mean ± standard deviation to improve statistical rigor.
4. We demonstrated generalization across model families by repeating our pipeline on LLaMA3-8B and observing similar large gains, and we added a vision-based variant with DinoV2 + scene-graph construction to show that our conclusions persist when scene graphs are recovered from images rather than given in text.
5. We clarified the exact scope of our EAI evaluation, making explicit that we focus on the Action Sequencing track and reporting the precise task counts per category. We also tempered and sharpened our claims to emphasize that the contribution is a high-level embodied decision-making training ground rather than an end-to-end perception-to-torque solution.
6. Finally, we corrected the template/formatting and typographical issues and incorporated the suggested related work (e.g., RoboVerse, Embodied World Models.) to better situate our framework in broader literatures.

We will incorporate these comments into the extended-page final version of our paper.

At the same time, we are grateful that all reviewers also highlighted several **strengths** of the work, which we briefly summarize for the AC:

1. Reviewers consistently note that the framework is well-engineered and that the pre-cached language–physics interface, resource scheduler, and task dispatcher work together to enable high-throughput rollouts, directly attacking a practical bottleneck in embodied RL. Ablations confirm that these components indeed accelerate simulation, which was explicitly described as “critical for RL practitioners.”
2. Conceptually, the work is recognized as tackling a central gap in embodied AI by proposing a unified training ground that connects language-only LLMs to physical, interactive decision-making, via multi-agent task generation, a scalable decoupled simulator, and a hierarchical reward curriculum.
3. Architecturally, reviewers appreciated the decoupled design, where the LLM trainer is separated from a distributed pool of heterogeneous simulation workers, enabling high-throughput parallel rollouts and improving the overall efficiency and scalability of the training loop.
4. The hierarchical reward design, especially the semantic relevance reward was noted as being carefully justified: the ablations and reward curves clearly show that this component is critical for guiding the agent in the early phase and enabling efficient, stable learning.
5. Multiple reviewers describe the paper as well-written, well-structured, and easy to follow, with clear figures and detailed explanations that make the technical contributions accessible despite the system’s complexity.

In summary, we view this work as a **first step toward a scalable, interactive training ground for embodied decision-making LLMs**, with a carefully engineered system and strong initial empirical results. We will incorporate these valuable comments into the extended-page final version of the paper. We hope that the clarifications, new experiments, and scope refinements in this rebuttal convincingly address the reviewers’ concerns, while preserving the strengths they identified regarding system design, empirical impact, and relevance to the embodied AI community.

---

### Meta-Review · Area_Chair_W17r · 2025-12-23

**Summary:**

The paper received divergent ratings from the reviewers (6,6,4,2). The reviewers appreciated strengths such as high-throughput rollouts, easy-to-read structure of the paper, outperforming larger models, and the data engine for LLM training. However, they raised several concerns as well:\
(a) Lacking essential details for data generation\
(b) Lack of confidence intervals for the results\
(c) Creating a self-referential loop when GPT is used for verification\
(d) Limited diversity due to using only 45 scenes\
(e) Strong claims about performance gains in EAI benchmarks\
(f) Lack of comparison with existing data generation methods\
(g) Loss of dynamic fidelity with pre-caching\
(h) The main results being only on 10 samples per task

The rebuttal addressed some of the concerns. For instance, it provided confidence intervals to address (b). To address (c), the authors provided results with human users and other LLMs (Grok-4, Gemini, and GPT-5). The AC read the paper, the reviews, and the rebuttal carefully. There are some major concerns that preclude acceptance: (1) Pre-caching the dynamics greatly oversimplifies the problem. What makes Embodied AI hard is the noise in actuation and perception. A good planner should take these into account. (2) The rebuttal provided comparisons with some prior work in data generation, but it misses PARTNR, ICLR 2025, which is very similar. PARTNR uses Habitat instead of the Behavior simulator, but the rest of this paper is very similar to PARTNR. PARTNR additionally considers perception and actuation noise and offers a more comprehensive approach to task generation and evaluation. (3) 10 samples per task are too few to draw any strong conclusions. There are other issues with this paper as well. Several important details are missing, such as the degree of overlap between the training and test sets, the rationale behind referring to the data generation method as ‘agents’, etc. In addition, the so-called ‘hierarchical’ reward formulation is not convincingly justified; it appears to be a straightforward addition of three reward terms rather than a true hierarchical structure. Due to these issues, rejection is recommended.

**Reviewer Concerns:**

Explained above.

**Reviewer Scores:**

Reviewer LYJB: No change would be expected since there are several missing details, and the rebuttal has not sufficiently addressed that.

Reviewer F41E: The concerns regarding data diversity still persists. So, no score change would be expected.

Reviewer SzXd: The rebuttal still misses comparisons with very similar prior work. Also, the criticism about the limited scope is valid. So, no score change is justified.

Reviewer 8Z5X: Pre-caching oversimplifies the problem. Addressing that is beyond the scope of the rebuttal, and it requires substantial experimentation. The reviewer would probably kept the current score.

---

### Decision · Program_Chairs · 2026-01-26

Reject